



# Noise filtering options for conically scanning Doppler LiDAR measurements with low pulse accumulation

Päschke Eileen[1] and Detring Carola[1]

[1]Deutscher Wetterdienst, Meteorological Observatory Lindenberg – Richard-Aßmann-Observatory, Lindenberg, Germany

**Correspondence:** Eileen Päschke (eileen.paeschke@dwd.de)

**Abstract.** Doppler lidar (DL) applications with a focus on turbulence measurements sometimes require measurement settings with a relatively small number of accumulated pulses per ray in order to achieve high sampling rates. Low pulse accumulation comes at the cost of the accuracy of radial velocity estimates and increases the probability of outliers, also referred to as "bad" estimates. Careful noise filtering is therefore the first important step in the data processing chain from raw data to the retrieved

turbulence variable. It is shown that commonly applied filtering techniques have weaknesses in distinguishing between "good" and "bad" estimates with the sensitivity needed for a turbulence retrieval. For that reason, new ways of noise filtering have been explored, taking into account that the DL background noise can differ from generally assumed white noise. It is shown that the introduction of a new coordinate frame for a graphical representation of DL radial velocities from conical scans offers a different perspective onto the data when compared to the well-known velocity-azimuth-display (VAD) and thus opens up new

possibilities for data analysis and filtering. This new way of displaying DL radial velocities builds on the use of a phase-space perspective. Following the mathematical formalism used to explain a harmonic oscillator, the VAD's sinusoidal representation of the DL radial velocities is transformed into a circular arrangement. Using this kind of representation of DL measurements, "bad" estimates can be identified in two different ways. Either in a direct way, by singular point detection in subsets of radial velocity data grouped in circular rings, or indirectly, by localizing circular rings with mostly "good" radial velocity estimates

by means of the autocorrelation function. The improved performance of the new filter techniques compared to conventional approaches is demonstrated, both through a direct comparison of unfiltered with filtered data sets, and through a comparison of retrieved turbulence variables with independent measurements.



## 1 Introduction

Doppler lidars (DL) are nowadays widely used for measurements of atmospheric wind and turbulence variables in different application areas, such as wind energy, aviation and meteorological research (Liu et al. (2019); Sathe and Mann (2013); Thobois (2019); Krishnamurthy et al. (2013); Filioglou et al.; Drew et al. (2013); O'Connor et al. (2010); Sathe and Mann (2013); Bodini et al. (2018); Sanchez Gomez et al. (2021); Beu and Landulfo (2022)). The wide application range became possible due to the flexible configuration options of several modern systems taking profit from the all-sky-scanner technique. This technique allows for the employment of user-defined scan patterns w.r.t. azimuth and elevation angles, and sampling frequencies specifically designed to meet the data needs for a specific application-oriented question.

At the Meteorological Observatory Lindenberg - Richard Aßmann Observatory (MOL-RAO) DL systems are primarily used to continuously measure vertical profiles of wind throughout the whole troposphere (Päschke et al., 2015) and to derive vertical profiles of various turbulence variables (e.g. eddy dissipation rates (EDR), turbulence kinetic energy (TKE), momentum fluxes) in the atmospheric boundary layer (ABL) with high temporal and vertical resolution. Such routine long-term observations of wind and turbulence variables are of interest in many respects. The data can be helpful in analyzing and interpreting the kinematic properties of the vertical structure of the atmospheric wind and turbulence under different weather conditions and states of the ABL during the course of the day (e.g. stable ABL, convective-mixed ABL, transitions between different ABL states). In addition, the profile information can be useful for regular validation purposes of atmospheric numerical models. This includes not only modeled wind profiles but also the performance of turbulence closure parameterizations (e.g. TKE closure) used to describe subgrid-scale processes. Similarly, new approaches to describe turbulence in numerical models are required, due to increasingly higher model resolution and the associated changes in the applicability and relative importance of parameterization schemes.

A variety of scanning techniques and retrieval methods for vertical profiles of wind and turbulence variables based on DL measurements have been developed (Smalikho (2003), Päschke et al. (2015), Sathe et al. (2015), Newsom et al. (2017), Bonin et al. (2017), Steinheuer et al. (2022)). Several of these methods rely on specific scanning configurations and are tailored towards a specific data product. For the derivation of different data products this implies either the use of more than one DL system or cyclic configuration changes of a single DL. With respect to this limitation, the relatively new scanning- and retrieval method introduced by Smalikho and Banakh (2017) stands out from other methods. Their approach is based on a carefully derived set of model equations, describing functional relationships between radial velocity observations measured along a conical scan with high azimuthal and temporal resolution ($\Delta\theta \sim 1°$, $\Delta t \sim 0.2$ s) and a set of meaningful wind turbulence variables such as TKE, EDR, momentum fluxes and the integral scale of turbulence. Hence, the essential benefit of this approach relies in the deployment of an internally consistent set of simultaneous wind and turbulence profile observations based on just one scan strategy. As a further outstanding feature the method provides additional correction terms to account for the typical underestimation of the TKE due to the averaging over the probe volume of the DL. This issue has frequently been mentioned as the most challenging task in turbulence measurements using DL (Sathe and Mann (2013); Liu et al. (2019)).



Because of the strength of the Smalikho and Banakh (2017) approach, the method has been implemented and tested for routine application at MOL-RAO. From first quasi routine test measurements with a StreamLine DL from the manufacturer HALO Photonics (nowadays HALO Photonics by Lumibird) three things became apparent: (1) the measurements of radial velocity show an increased level of noise which is noticeable through an increased number of outliers ("bad" estimates) even at rather low height levels in the ABL, (2) the accuracy of retrieved wind and turbulence variables strongly depends on the quality of the input data and (3) if the signal-to-noise-ratio (SNR) thresholding technique is used to remove noise from the data, the final turbulence product availability is relatively low. The first finding reflects the basic disadvantage of the high spatio-temporal scanning strategy which can be understood as follows: The DL transmits short pulses of infrared laser light of wavelength $\lambda$ into the atmosphere where they interact with aerosols carried by the wind. The bulk of all aerosols within the illuminated pulse volume act as backscattering targets. The wind-induced motion of the aerosol particles cause a frequency shift $f_d$ in the spectra of the backscattered light which provides information on the radial wind velocity through $V_r = -\lambda f_d/2$, i.e., the projection of the wind vector onto the measurement direction. The longer a signal is sampled, the more accurate the measurement will be. For that reason it is a common approach to accumulate the spectra of backscattered light from multiple pulses $N_a$ (Frehlich, 1995; Rye and Hardesty, 1993; Banakh and Werner, 2005; Li et al., 2012). For the retrieval of wind profiles as proposed in Päschke et al. (2015), for instance, DL measurements have been performed using a comparably high number of $N_a = 75000$ pulses. At this point the method of Smalikho and Banakh (2017) requires a sensible compromise. Using a StreamLine DL, a conical scan with the required high azimuthal and temporal resolution can only be achieved with a rather low number of accumulated pulses per measurement ray, i.e. $N_a \sim 2000$. This in turn has the consequence that the occurrence of "bad" estimates in the measurements becomes more likely (Frehlich, 1995). Such outliers contain no wind information (Wildmann and Smalikho, 2018) and, if not excluded from the measured data set, they may contribute to large errors in the retrieved meteorological variables (Dabas, 1999). Latter explains the aforementioned second finding and indirectly confirms the recommendations given in Banakh et al. (2021) that the method for determining wind turbulence parameters according to the work of Smalikho and Banakh (2017) is only applicable, if the probability $P_b$ of "bad" estimates of the radial velocity is close to zero. A closer examination of the third finding mentioned above revealed that with the proper choice of the threshold value the SNR thresholding technique is indeed very effective in removing noisy data, but it also bears the risk to discard a lot of reliable measurements. This in turn proves to be ineffective for the overall product availability and would not justify a routine application of the retrieval method.

As described above, first test measurements revealed unexpected difficulties with noise contaminated measurements during a 24/7 application and therewith varying atmospheric conditions w.r.t. the occurrence of backscattering targets. This gave us the motivation to think about ideas for new filtering techniques which allow for a reliable removal of all noise contributions but circumvent an unnecessary refusal of reliable data at the same time. A detailed presentation of these ideas is the main objective of this work. The article is organized as follows: In Sect. 2 we describe the DL system used during our measurement program and discuss examples showing typical characteristics of the quality of measurements based on short sampling times because of low $N_a$. In Sect. 3 we discuss the limitations of common filter methods to detect "bad" estimates in order to motivate our efforts to develop new ideas for improved filtering techniques. In Sect. 4 ideas for two new filtering techniques are presented,





which are based on the use of the phase-space perspective for DL radial velocity representation from conically scanning DL systems. In Sect. 5 the advantages and disadvantages of the new filtering techniques and the consequences for their operational implementation are discussed by means of typical measurement examples.

## 2 Measurements

This section provides information concerning the type of the DL system used for the measurement and gives insights into typical measurement results as obtained by applying the measurement strategy suggested in Smalikho and Banakh (2017). The focus is on different types of noise distributions that have been observed. Additionally we show that the autocorrelation function (ACF) of the time series of radial velocity measurements from conical scans can be used as a noise indicator that shows whether the series in its entirety is noisy or not. This idea is closely related to the work of Frehlich (2001), who used the autocorrelation to estimate the velocity error of DL measurements. In Sect. 4.2.2 this noise indicator property of the ACF will be exploited for a filtering method that separates noise from reliable measurements.

### 2.1 System specification and configuration

At MOL-RAO a StreamLine DL from the manufacturer HALO Photonics with the specifications given in Table 1 is used for ABL wind and turbulence measurements. The conical scan mode, adopted from Smalikho and Banakh (2017), is defined by three key parameters, namely the elevation angle ($\phi = 35.3°$), the azimuthal resolution ($\Delta\theta \sim 1°$) and the time duration for one single scan ($T_{scan} = 72$ s). In order to realize this scanning strategy the DL was configured to be in continous scan motion (CSM) while sampling data. Because of the time-consuming stops at pre-defined waypoints (i.e. pre-defined azimuth angles along the scan circle) for the StreamLine DL systems, a step-stare scan configuration appeared not practicable. A custom scan file (see Appendix A) has been defined for the scanner configuration including information about the angular rotation rate $\omega_s$, the start/end positions of the scanner and the elevation angle $\phi$. In analogy to the work of Smalikho and Banakh (2017) we set $\omega_s = 5°$ s$^{-1}$ to nearly satisfy $\Delta\theta \sim 1°$. Note that the latter implies measurements on an irregular grid which for analysis purposes later on requires the transfer of the data to an equidistant grid with $\Delta\theta = 1°$. The specific value for $\phi$ goes back to an earlier theoretical work of Kropfli (1986) and Eberhard et al. (1989) with focus on Doppler radar based turbulence measurements. In addition, Teschke and Lehmann (2017) have recently shown that using DL this value is also an optimum beam elevation angle for a mean wind retrieval with a minimum in the retrieval error. With the specifications for $\Delta\theta$, $\omega_s$ and due to the pulse repetition frequency $f_p = 10$ kHz (see Table 1) we had to adjust the configuration setting for the number of pulses per ray to $N_a = 2000$, using the relation $N_a = \Delta\theta f_p/\omega_s$ (Banakh and Smalikho, 2013). This is a minor difference compared to the value suggested in Smalikho and Banakh (2017), i.e. $N_a = 3000$, which is due to a higher pulse repetition frequency, i.e. $f_p = 15$ kHz, characterizing their DL system. Note that for StreamLine DL systems the system specific parameter $f_p$ cannot be changed by the user. The low value for $N_a$ is non-favourable if a high measurement quality is needed. For best possible measurement quality in the lower ABL it is therefore important to use the focus setting option to improve the signal intensities within a selected height range. For the DL used in our studies (DL78 hereafter) the focus was set





to 500m. Working with StreamLine DL systems, also the range resolution $\Delta R$ along the line of sight (LOS) can be adjusted. For reasons of compatibility with the pulse length of $\tau_p = 180$ ns the range resolution was set to $\Delta R = c\,\tau_p/2 \approx 30$ m, where
$c$ denotes the speed of light.

Note that with StreamLine XR DL systems in the meantime HALO Photonics by Lumibird offers a further development of the StreamLine series. XR systems operate with larger pulse length in order to increase the range, depending on the presence of scattering particles in the atmosphere. The larger pulse length, however, reduces the spatial resolution of the measurements along the line-of-sight (LOS) which is no option for measurements in the ABL, if the focus is on the detection and investigation
of small-scale structures.

**Table (1).** Instrument specifications of the HALO Photonics StreamLine DL operated at MOL - RAO.

| Instrument specifications | |
| --- | --- |
| Serial number | 0414-78 |
| Wavelength | 1.5 $\mu$m |
| Pulse length | 180 ns |
| Pulse repetition frequency | 10 kHz |
| Sampling frequency | 50 MHz |
| Maximum range | 7.5 km |
| Bandwidth | $\pm 19.4$ m/s |

## 2.2    Characteristic measurement examples

There are a number of estimation algorithms to determine the frequency shift $f_d$ and therewith the radial velocity $V_r \propto f_d$ from the spectra of backscattered light. A selection of some algorithms is discussed in Frehlich (1995). The performance of parameter estimators can vary widely. Generally, the assessment of the performance of the estimation algorithms is based on
the probability density function (PDF) of the velocity estimates taking into account instrumental parameters characterizing the measurement system. According to Frehlich (1995), the PDF of velocity estimators performing well is characterized by a localized distribution of "good" estimates centered around the true mean velocity and a fraction of uniformly distributed "bad" estimates or random outliers. This leads to the following frequent distinction in the DL radial wind measurements:

$$V_r = \begin{cases} \mathsf{V}_r + \mathsf{V}_e & \text{in case of "good" estimate} \\ \mathsf{V}_b & \text{in case of "bad" estimate} \end{cases} , \tag{1}$$

whereas $\mathsf{V}_e$ denotes a random instrumental error (Wildmann and Smalikho, 2018).

A uniform distribution of "bad" estimates indicates that the noise component of the spectrum of the lidar signal is white noise (Wildmann and Smalikho, 2018). In literature, "bad" estimates are mostly described as random outliers that are uniformly distributed over the resolved velocity space (Frehlich, 1995; Dabas, 1999). However, from our user experience with DL measurements we know that "bad" estimates do not always have to be uniformly distributed. Having in mind that the total





noise component in DL measurements can have different sources (e.g. shot noise, detector noise, relative intensity noise (RIN), speckles) (Hellhammer, 2018), whereby not all these sources represent white noise, we hypothesize that the total DL noise must not necessarily be white noise and depends on the strength of each of the above listed different types of noise.

In the following subsection we show examples of noise-contaminated measurements obtained with a HALO StreamLine working with a scanning strategy as described above. The focus is on the phenomenological description of different noise

categories.

### 2.2.1 Categories of distributions of "bad" estimates

Three typical examples for DL78 measurements are shown in Fig. 1 which differ in terms of their noise characteristics. To get comprehensive insights into the properties of the 30-minute measurements, different analysis diagrams of each measurement example are summarized column by column. Noise free measurements are shown in the left column. Apart from the superim-

posed small-scale fluctuations which reflect natural turbulent fluctuations, this can be seen in the undisturbed sinusoidal course in the time series plot of the radial velocities (Fig. 1[a]) and the corresponding VAD plot (Fig. 1[c]). The sinusoidal course is typical for measurements from conically scanning DL systems during stationary wind field conditions and manifests itself in a U-shaped bimodal distribution of the radial velocities shown in the histogram (Fig. 1 [d]). In the middle and right columns the measurements are contaminated with noise. Here, the periodic signal is temporarily interrupted by "bad" estimates randomly

representing "any" value in the velocity range $\pm 19$ m s$^{-1}$ (Fig. 1 [e], [i], [g], [k]). Furthermore, differences in the distribution of "bad" estimates are noticeable. In contrast to the measurements in the middle column where the "bad" estimates appear quite uniformly distributed (type A noise hereafter), an additional higher aggregation of "bad" estimates around zero (type B noise hereafter) is noticeable in the right column. This becomes particularly clear by comparing the corresponding subfigures with the VAD (velocity azimuth display) diagrams (Fig. 1 [g],[k]) and those with the histograms of the radial velocities (Fig.

1 [h],[l]). Note that contrary to Fig. 1 [d] the characteristic U-shaped distribution in Fig. 1 [l] can no longer be recognized because it mixes with a Gaussian-like distribution of "bad" estimates. Finally, for each measurement example a clear difference in the level of the signal intensities is noticeable (Fig. 1 [b], [f], [j]). With SNR values around -10 dB the signals are strong in the noise free case and with values smaller than -15 dB the signals are weak in the noisy cases. It is important to point out that for the noisy cases the signal levels are mostly the same and thus do not provide any indication about the type of noise

distribution.

To the authors' knowledge, up to now there are no user reports about "bad" estimate distributions of type B in DL measurements available. This can possibly be explained by the fact that end users with interests in meteorological applications mostly work with high numbers of pulse accumulation in order to achieve the best possible signal quality. We can also rule out that the occurrence of type B noise is a system specific DL78 problem. During the FESSTVaL (Field Experiment on Sub-

mesoscale Spatio-Temporal Variability in Lindenberg) campaign (Hohenegger et al., 2023), we had the opportunity to compare the measured data of four StreamLine and four StreamLine XR Doppler Lidars (see Sect. 2.1) positioned side by side and configured identically using the scan mode outlined in Smalikho and Banakh (2017). The comparison revealed type B like noise contamination within the measured data for several systems albeit in varying degree.







**Figure (1).** Examples for measurements from one and the same conically scanning Doppler lidar. Each column represents measurements during a 30 min interval at different times and range gates (i.e. measurement heights along the line of sight) which are characterized by different kinds of noise (left: noise free, middle: type A noise, right: type B noise). The plots of each row depict the measurements from different perspectives. The first row shows a time series plot of the radial velocities ($V_r$) ([a], [e], [i]). In a similar way the second row illustrates the corresponding signal intensities (SNR) of the measurements. Here, the horizontal dotted line indicates an SNR - threshold level calculated as proposed in Abdelazim et al. (2016) for $N_a = 2000$ (see Sect. 3.1.1). The third row shows the DL measurements from a VAD perspective, i.e. a display of the radial velocity as a function of the azimuth angle. The ACF value indicates the degree of noise contamination in the measured time series (see Sect. 2.2.2). The fourth row shows histograms of $V_r$.





### 2.2.2 Quantification of the occurrence of noise

A quantification of the occurrence of noise in a series of radial velocity measurements based on a conical scan is feasible by means of the ACF. Noise-free radial velocity measurements based on a conical scan geometry follow a sinusoidal curve when plotting $V_r$ against its corresponding azimuthal measurement direction $\theta$. Hence, consecutive measurements along the curve are not independent from each other and with a high azimuthal resolution of the measurements neighboring radial velocities are highly correlated. The situation is different with noise contaminated measurements. Here, the sinusoidal course in the series

of consecutive measurements is occasionally interrupted by lying far-off "bad" estimates such that consecutive measurement values are completely independent of each other, i.e., they do not correlate. An indication to what degree a signal is similar to a shifted version of itself as a function of the displacement $\tau$ is given by the ACF. It will be shown next that the ACF proves to be helpful to distinguish between measurement periods with and without noise contamination.

Assuming horizontally homogenoeus and stationary wind field conditions, DL measurements taken along the scanning

circle are described through $V_r = u\,sin(\theta)\,sin(\phi) + v\,cos(\theta)\,sin(\phi) + w\,cos(\phi)$ , where $u, v, w$ denote the 3D wind vector components, $\theta$ the azimuth angle determining the measurement direction along the scanning circle and $\phi$ denotes the zenith angle of the inclined laser beam (Päschke et al., 2015). Note that for a continous conical scan over a number of rounds the azimuthal angle in turn has to be regarded as a function of time, i.e. $\theta = \theta(t)$. This makes the DL measurements over a specified time interval a periodic signal, reading:

$$V_r = sin(\phi)\left[u\,sin\left(\frac{2\pi}{T_{scan}}t + t_0\right) + v\,cos\left(\frac{2\pi}{T_{scan}}t + t_0\right)\right] + w\,cos(\phi) \qquad . \tag{2}$$

For simplicity let's assume $v = 0$. Then it can be shown that the ACF of the periodic signal $V_r(t)$ has a cosine shape, i.e.

$$ACF(\tau^*) = \frac{u^2 sin^2(\phi)}{2}cos\left(\frac{2\pi}{T_{scan}}\tau^*\right) + w^2 cos^2(\phi) \qquad , \tag{3}$$

where $\tau^*$ denotes the time delay (Sienkowski and Kawecka, 2013). In case of returning to a view of $V_r$ as a function of the azimuthal angle $\theta$ and taking into account that in the atmosphere vertical motions are typically much smaller than horizontal

ones, i.e. $w \ll u$, Eq. (3) can be further simplified and rewritten in the following normalized form

$$\frac{ACF(\tau)}{C} = cos(\tau) \qquad , \tag{4}$$

with $C = 2u^{-2}sin^{-2}(\phi)$. Using this notation, the parameter $\tau = (2\pi/T_{scan})\tau^\star$ denotes an azimuthal displacement. For discrete and equidistant measurements along the scan circle latter generally can be written as $\tau = n\,\Delta\theta$ where $\Delta\theta$ denotes the measurement resolution along the scanning circle and $n \in \mathbb{N}$ denotes the lag index of the displacements in the series of

measurements. Hence, using Eq. (4) it can be easily verified that for noise-free DL measurements with an azimuthal resolution of $\Delta\theta = 1$ deg the correlation between neighboring measurements, i.e. $n = 1$, along the scan circle is close to one, i.e. ACF($\tau = 1$) $\approx 1$ (here, $\Delta\theta$ is in radians). Note, however, that with decreasing azimuthal measurement resolution, i.e. an increase in $\Delta\theta$, the distance between neighboring measurements becomes larger and thus the correlation lower. For instance, with an azimuthal measurement resolution of $\Delta\theta = 36$ deg a correlation of ACF($\tau = 36$) $\approx 0.8$ between neighbouring points



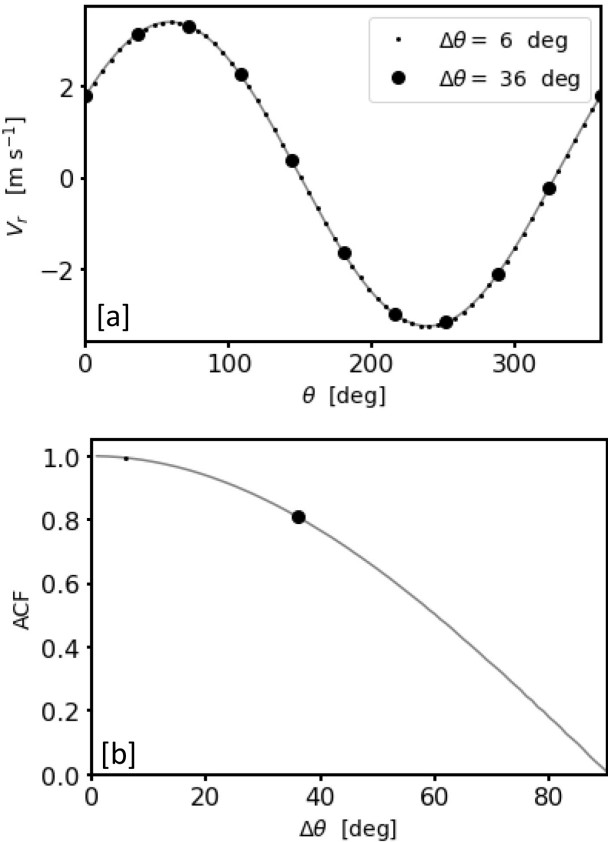

**Figure (2).** [a] Two theoretical examples for radial velocity measurements $V_r$ from a conically scanning Doppler lidar with different azimuthal resolution, i.e. $\Delta\theta = 6°$ and $\Delta\theta = 36°$. Grey-colored in the background the corresponding continuously resolved sine curve. [b] Normalized lag k = $\Delta\theta$ autocorrelation values of the continous sine shown in [a] for azimuth steps between $\Delta\theta = 1°$ to $\Delta\theta = 90°$. The dots indicate the maximum expected value of the autocorrelation function ACF (n = 1; $\Delta\theta$) = ACF$_{max}$ between neighboring azimuthal measurement points for the discrete measurement series of different azimuthal resolution $\Delta\theta$ shown in [a].

may be expected (Fig. 2). These theoretically derived results can be verified by calculating the ACF for the noise-free measurements shown in Fig. 1. Note that despite the obvious non-inhomogeneity and non-stationary of the wind field the correlation of neighboring measurement points is close to 1, i.e. ACF($\tau = 1$) $\approx 1$ (see Fig. 1 [c]). In contrast, by considering the noise contaminated measurement examples the occasional independence of neighboring measurements due to the occurrence of "bad" estimates interrupting the sinusoidal course of the measurements is reflected by low correlation values, namely ACF($\tau = 1$)

$\approx 0.77$ for the type A noise example and ACF($\tau = 1$) $\approx 0.33$ for the type B noise example (see Fig. 1 [g], [k]). Hence, these examples show that the ACF can be used as an indicator for the general occurrence of noise in the series of DL measurements,



however, without precise knowledge about which of the measurements are "bad" estimates. Generally it holds:

$$\mathrm{ACF}(n = 1; \Delta\theta) = \mathrm{ACF}_{max} \quad : \quad \text{no noise}$$

$$\mathrm{ACF}(n = 1; \Delta\theta) < \mathrm{ACF}_{max} \quad : \quad \text{noise} \qquad .$$

Here, $\mathrm{ACF}_{max}$ determines the maximum expected correlation between neighboring measurements points indicated by $n = 1$. Depending on the azimuthal resolution $\Delta\theta$, $\mathrm{ACF}_{max}$ takes different values which can be calculated using Eqn. (4).

## 3  Turbulence products based on common filtering techniques

In the previous section it has been shown that DL radial velocity measurements obtained using the measurement strategy proposed in Smalikho and Banakh (2017) can show a strikingly high proportion of noise, if the conditions for measurements
are unfavourable. A successful application of their method to determine turbulence variables, however, requires a probability of "bad" estimates close to zero (Smalikho and Banakh, 2017; Banakh et al., 2021). Hence, a careful pre-processing to detect and remove noise from measured data sets is necessary. This section intends to draw attention to potential emerging issues if the pre-processing of DL measurements does not properly remove "bad" estimates. For that purpose a short overview about frequently used filtering techniques will be given. Its strengths and weaknesses are discussed in different ways. On the one hand,
we consider selected measurement examples and directly compare the original series for $V_{meas}$ with the immediate outcome of the filtering techniques. On the other hand, we employ a comparison of turbulence products derived from differently pre-filtered DL data with independent sonic measurements. Since it is expected that residual noise or data which despite valid measurement information mistakenly have been removed from the measured raw data set can contribute to errors in the retrieved turbulence variables, the comparison of retrieved turbulence variables can be used as an indirect method to evaluate the performance of
different filtering techniques.

### 3.1  Commonly used filtering techniques

Meanwhile, with the increased interest in DL measurements for meteorological applications, different filtering techniques to separate reliable data from noisy measurements can be found in literature. A closer look at the underlying principles of quality assessment of radial velocity estimates allows a rough subdivision into two categories of filtering methods: (1) One category
makes use of additional range resolved measurements provided by the instrument and (2) the other uses statistical analysis tools applied to time series of DL radial velocity estimates. The method behind the first category is the well known SNR thresholding technique. Methods representing the second category are for instance the Median Absolute Deviation (MAD) originating from Gauss (1816), the Consensus Averaging (CNS) introduced by Strauch et al. (1984), the filtered-sine-wave-fit (FSWF) by Smalikho (2003) or the integrated iterative filter approach by Steinheuer et al. (2022). The last two methods
mentioned are directly integrated into a retrieval method for wind or wind gusts. A review of further filtering methods that belong to the second category mentioned above is given in Beck and Kühn (2017). In the following we discuss in more detail the advantages and disadvantages of the different filter categories using the examples of the SNR thresholding and CNS





techniques, respectively. It is important to point out that the discussion concentrates on their applicability as a pre-processing
step prior to a turbulence product retrieval if the DL radial velocity measurements were carried out with low pulse accumulation
(e.g. $N_a = 2000$).

### 3.1.1   SNR-thresholding

Starting from DL raw data, the SNR is determined from signal spectra which are calculated in the further course of the signal
processing used in DL technology. It is defined as the ratio between the signal power to the noise power. The first bears the
meaningful information in a measurement and the latter is considered as an unwanted signal contribution that is blurring this
information. The higher the level of the signal power and the smaller the level of the noise power, the better the SNR and thus
the quality of the radial velocity estimate. Note that under unfavourable measurement conditions, e.g. weak backscattering
from the atmosphere, the challenge relies in a clear distinction between signal and noise in the spectra. Here, the method of
pulse accumulation is a useful tool to get clearer results (Frehlich, 1996) and thus helps to improve the quality of the DL radial
estimates.

For DL users with interest in automated routine applications for long-term observations, knowledge about both a suitable
SNR-threshold ($SNR_{thresh}$, hereafter) to separate "good" from "bad" estimates and an error estimate for the radial velocity
would be desirable. Pearson et al. (2009) provides a guideline on that issue based on an experimental approach. His results
showed good agreement with theoretical results based on an approximate equation introduced by Rye and Hardesty (1993),
reading:

$$\sigma = 2 \, (\pi^{0.5}/\alpha)^{0.5} \, (1 + 0.16\alpha) \, (\Delta\nu / N_p^{0.5}) \quad , \tag{5}$$

with $\alpha = SNR/((2\pi)^{0.5}(\Delta\nu/B))$ and $N_p = M \, N_a \, (SNR)$. Here, $\sigma$ denotes the error estimate of the radial velocity in the
weak signal, multipulse-averaged regime, $N_a$ the number of accumulated pulses, B the bandwidth, $\Delta\nu$ the signal spectral
width and M the gate length in points. For more details see Appendix B. Note that Eq. (5) can be used in two ways. On the one
hand, it provides an estimate for the uncertainty of the radial velocity estimate depending on the SNR. On the other hand, it
provides guidance to calculate $SNR_{thresh}$ for a prescribed acceptable uncertainty on the Doppler lidar estimate. Examples for
an evaluation of Eqn. 5 for different numbers of $N_a$ are given in Fig. 3. The curves basically show how the uncertainty of the
measurements decreases with increasing SNR. Additionally, the effect of pulse accumulation becomes visible. For the same
requirement on the uncertainty of the Doppler estimate, e.g. $\sigma < 0.5$ m s$^{-1}$ or $\sigma < 0.1$ m s$^{-1}$, the corresponding SNR-threshold
value for reliable data would be lower for Doppler estimates based on higher pulse accumulations (e.g. $SNR_{thresh}$ = -24 dB
or $SNR_{thresh}$ = -17 dB for $N_a = 30000$) than for Doppler estimates based on a lower number of pulse accumulations (e.g.
$SNR_{thresh}$ = -18.5 dB or $SNR_{thresh}$ = -11 dB for $N_a = 2000$). A more simpler approximate equation to determine $SNR_{thresh}$
is suggested in Abdelazim et al. (2016). Here an equation has been derived taking into account the number of accumulated
pulses $N_a$ only and which reads:

$$SNR_{thresh} = \frac{1}{\sqrt{N_a}} + \frac{\sqrt{2}}{\sqrt{N_a}} \quad . \tag{6}$$





| | $N_a$ = 2000 | $N_a$ = 3000 | $N_a$ = 30000 |
|---|---|---|---|
| **Abdelazim et al. (2016)** | -12.7 dB | -13.55 dB | -18.5 dB |
| **Rye and Hardesty (1993)** for σ < 0.5m/s : | -18.5 dB | -19.6 dB | -24.0 dB |
| for σ < 0.1m/s : | -11.0 dB | -12.0 dB | -17.2 dB |

**Figure (3).** top: Examples of calculated SNR-threshold values depending on the number of accumulated pulses $N_a$ based on the approach by Abdelazim et al. (2016) and the approach by Rye and Hardesty (1993) . bottom: Example plots for the change of the theoretical standard deviation $\sigma$ of the Doppler velocity estimates depending on the signal-to-noise ratio (SNR) following the approach by Rye and Hardesty (1993). The curves are valid for different $N_a$ and the following system specific parameters: B = 2*19 m s$^{-1}$, M = 10, $\Delta\nu = 1.3\ m\ s^{-1}$.

Example results for SNR$_{thresh}$ derived from Eq. (6) are also given in Fig. 3.

  The retrieval of turbulence variables requires a high quality of the DL measurements. This raises the question how the uncertainty requirement on the DL estimate has to be chosen in order to exclude any "bad" estimates. To answer this question we applied the SNR threshold examples summarized in Fig. 3 to real measurements which are shown in Fig. 4. Here, estimates for $V_r$ are displayed against their associated SNR values. The measurement examples are obtained from the same DL system

but are based on different types of conically scanning modes working with different numbers of pulse accumulations $N_a$. Note that the plots are also valid for different days, i.e. the measurements were obtained under different atmospheric conditions concerning wind and ABL aerosol content. When comparing all three subfigures of Fig. 4 with each other it can be clearly seen that the SNR range where "bad" estimates randomly fill the entire search band ± 19 m s$^{-1}$ decreases with increasing pulse accumulation numbers. Latter is most evident at the upper and lower edges of the search bands. Seeking for a useful

SNR threshold to clearly distinguish between "good" and "bad" estimates, the approach by Abdelazim et al. (2016) turns out to be most appropriate. This only seems possible with high demands on the uncertainty of the measurement, i.e. $\sigma < 0.1$  m s$^{-1}$, which implies a comparably conservative high SNR threshold, i.e. $SNR_{thresh} = -12.7$  dB for $N_a = 2000$. Otherwise a huge fraction of "bad" estimates would remain in the data set. It is worth to point out that this SNR$_{thresh}$ value is nearly





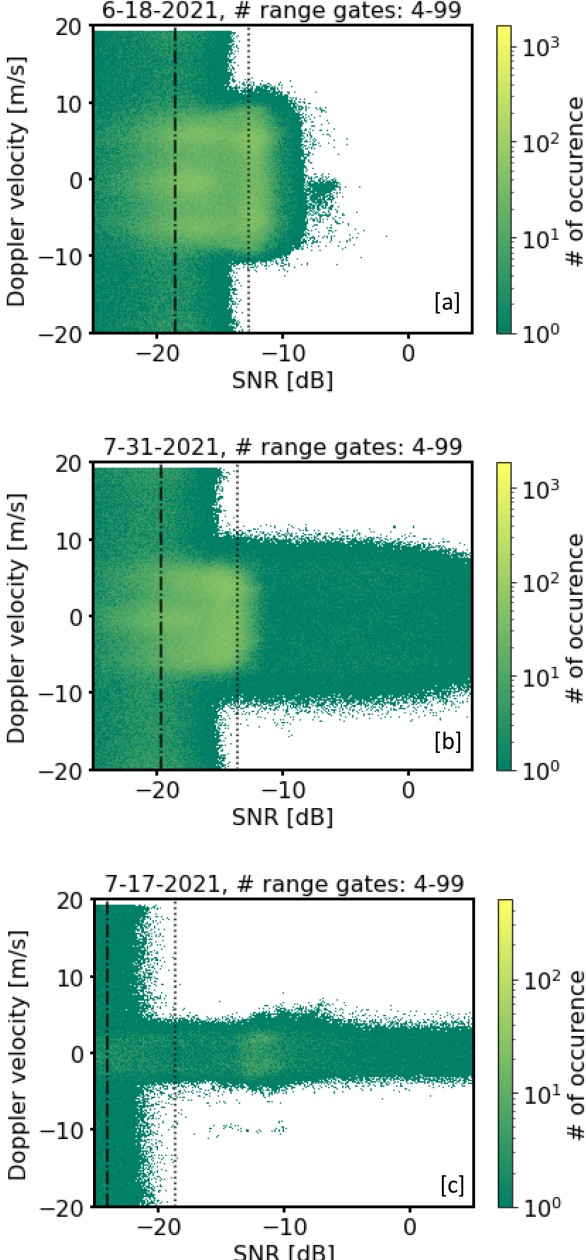

**Figure (4).** Doppler velocity vs. SNR plots for three different DL measurement examples. All measurements were taken using a conical scan but differ in the number of accumulated pulses N ([a]: $N_a$ = 2000, [b]: $N_a$ = 3000, [c]: $N_a$ = 30000). Each plot includes full day measurements for all range gates between the 4th and 99th range gate. The vertical lines in each plots denote different SNR-thresholds based on different approaches, namely by Abdelazim et al. (2016) with an Doppler velocity uncertainty of $\sigma < 0.1\ m\ s^{-1}$ (dot) and by Rye and Hardesty (1993) with an Doppler velocity uncertainty of $\sigma < 0.5\ m\ s^{-1}$ (dash dot). See Fig. 3 for exact SNR-threshold numbers.





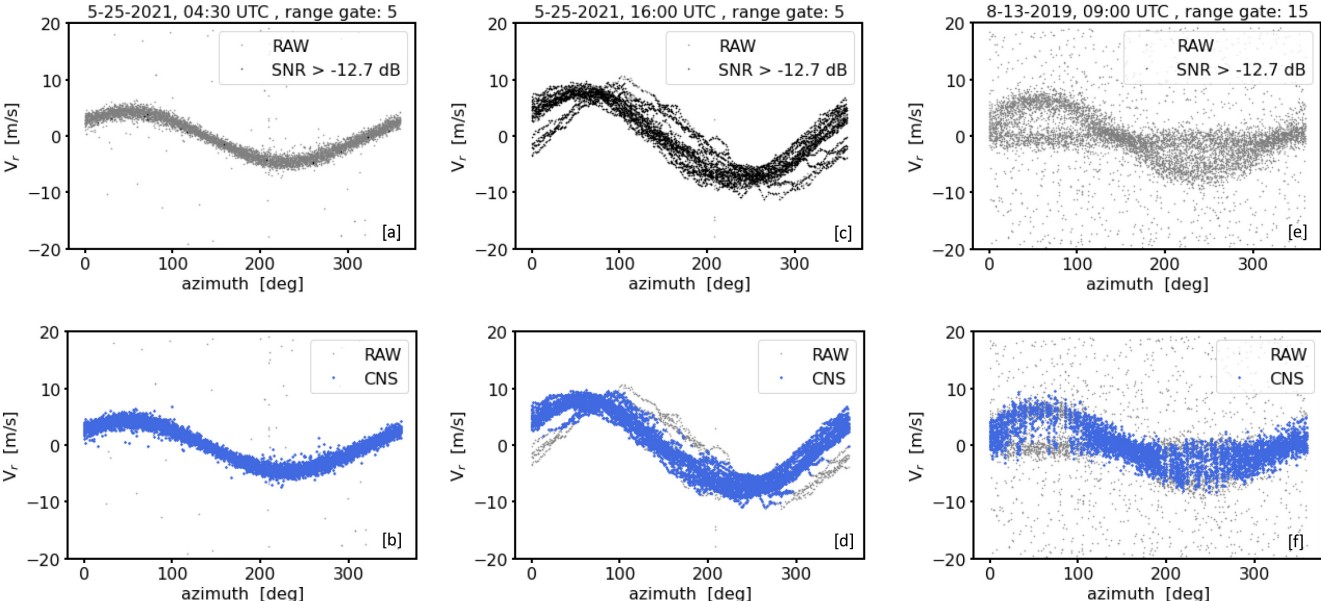

**Figure (5).** VAD plot examples from conical DL measurements with $N_a$ = 2000 pulse accumulations illustrating the application of the SNR thresholding ([a],[c],[e]) and the CNS ([b],[d],[f]) noise filtering method. The examples represent three different 30min measuring intervals at different range gates and with different level of noise contamination. The SNR - threshold value of -12.7dB has been calculated using the approach by Abdelazim et al. (2016). $\Delta\theta = 1°$ and $\Delta V_r$ = 3 m s$^{-1}$ was used for the application of the CNS.

in accordance with the one given in Banakh et al. (2021). Here, same conical scan configurations as proposed in Smalikho
and Banakh (2017) has been used to estimate the mixing layer height based on measurements of the eddy dissipation rate. In this context, Banakh et al. (2021) pointed out, that the SNR should be not less than -16 dB, when the relative error of lidar estimation of the dissipation rate should not exceed 30 %.

Unfortunately, by applying $SNR_{thresh} = -12.7$ dB to DL radial velocity measurements as shown with the examples given in Fig. 1 as well as Fig. 5 ([a],[c],[e]), a huge fraction of obviously "good" estimates would be discarded. A reduction of data
availability would be the consequence making a representative derivation of wind and turbulence products difficult or even impossible. This is a limiting factor of this kind of quality control (QC) procedure (Dabas, 1999).

### 3.1.2 Consensus averaging

Methodically different from the SNR thresholding technique is the consensus averaging (CNS) introduced by Strauch et al. (1984). The method was originally developed to exclude outliers from radar wind profiler data. A schematic that explains the
CNS approach is shown in Fig. 6. Here noise contaminated measurements of $V_r$ from several single conical scans executed one after the other are displayed using the VAD perspective. Separating the range of measurement directions (0° to 360°) into equidistant intervals $I_i$ = [(i-1) $\Delta\theta$, i $\Delta\theta$] (i = 1,2,3, ..., n with n $\in \mathbb{N}$), the basic idea is to seek within each $I_i$ along the $V_r$ axis for this subset of data satisfying both (i) the occurrence within a prescribed interval $\Delta V_r$ which is assumed to be a typical





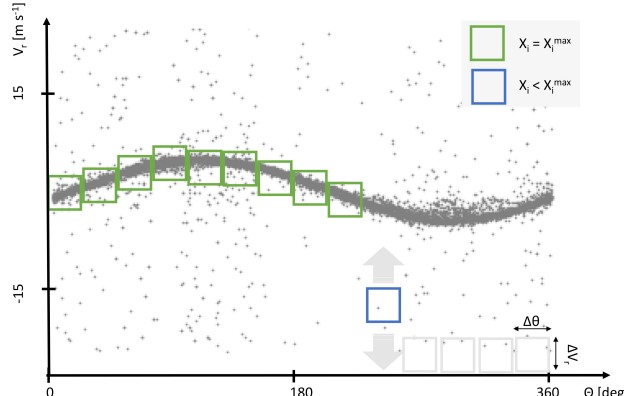

**Figure (6).** Schematic representation of a possible practical implementation of the CNS (consensus) averaging method, based on an approach by Strauch et al. (1984) (see Sect. 3.1.2). The data in the green boxes have already been identified as reliable. The blue box illustrates the process of searching the reliable data and the grey boxes stand for azimuthal intervals that still need to be analyzed. .

value for the atmospheric wind variance and (ii) the provision of a data availability $X_i^{max}$ which, however, must not fall below

a prescribed value $X_{thresh}$. Similar to the SNR-thresholding technique, the difficulty exists in a meaningful choice of $\Delta V_r$ and $X_{thresh}$ as will be shown in more detail below.

    If the focus is on the derivation of turbulence variables, including the determination of variances caused by eddies in turbulent flow, the problem of this CNS approach is that it requires an a-priori estimate of the variance which is actually being attempted to be derived. If $\Delta V_r$ does not correspond to the true atmospheric situation, e.g. the assumed value for $\Delta V_r$ is too small

or too large, it may happen that either measurements bearing relevant wind information are rejected or that "bad" estimates remain in the data set. Examples of this are given in Fig. 5 assuming $\Delta V_r = 3$ m s$^{-1}$ and $X_{thresh} = 60\%$. The early morning example shows noise contaminated DL measurements during weak wind and turbulence conditions (Fig. 5 [b]). Here, the actual wind variance was obviously lower than assumed by $\Delta V_r$, so that the prescribed interval $\Delta V_r$ gave room for the inclusion of "bad" estimates which had to be accepted as reliable due to the CNS concept. The afternoon example (Fig. 5 [d]) shows DL

measurements during stronger wind and turbulence conditions than in the morning. Additionally, at some point during the 30 min interval a change in the wind direction contributes to a phase shift in the sine signal represented by some of the scan circles (see Sect. 2.1). Mainly because of this non-stationarity the variability of the Doppler velocity measurements is obviously larger than assumed by $\Delta V_r = 3$ m s$^{-1}$ in some azimuth sectors so that relevant information characterizing this non-stationarity remains outside of the interval $\Delta V_r$ and are discarded by the CNS. Note that at this point the focus is on the performance

of the CNS and not on reconstructed wind and turbulence variables. Hence, this non-applicability of the CNS method during non-stationary measuring intervals needs to be considered apart from the question whether a derivation of wind and turbulence variables is meaningful if non-stationarity occurs. In Sect. 3.3 it will be discussed in more detail, that a wrong inclusion of "bad" estimates or a false exclusion of "good" estimates because of a non-compatible $\Delta V_r$ compared to the actual atmospheric



situation has the consequence that turbulence products (e.g. TKE) calculated based on CNS pre-filtered measurement data may

be either over- or underestimated.

Another limitation of the CNS filtering technique is that it expects a uniform distribution of "bad" estimates for a successful application. This becomes evident by considering the CNS filtering results for the measurement example shown in Fig. 5 [f]. This example represents the type B noise case shown in Fig. 1. Here, the subsets of Doppler velocities found by the CNS for some of the azimuthal sectors often do not represent the desired "good" estimates because the high density of "bad" estimates

around zero erroneously shifts the range $\Delta V_r$ bearing a maximum of data availability towards zero. In this case the non-uniform distribution of the "bad" estimates makes an successful application of the CNS impossible. Note, that the disadvantages worked out here can be generalized to all statistical methods used for outlier detection which require additional assumptions about the distribution and the variance of the quantity of interest.

## 3.2 TKE products using noise free measurements

For an indirect evaluation of the performance of different filtering techniques w.r.t. the derived turbulence variables we compare the TKE from differently pre-filtered DL measurements for a given measurement period in Sect. 3.3. Before, however, we have to ensure that the used retrieval method is suitable to generally give reliable results. This will be done using an example for one day (i.e. 04-04-2019) of measurements under favourable atmospheric backscatter conditions, characterized by high SNR values representing almost noise-free measurements, i.e. $V_r = \mathsf{V}_r + \mathsf{V}_e$. Evidence for this can be found in Fig. 7.

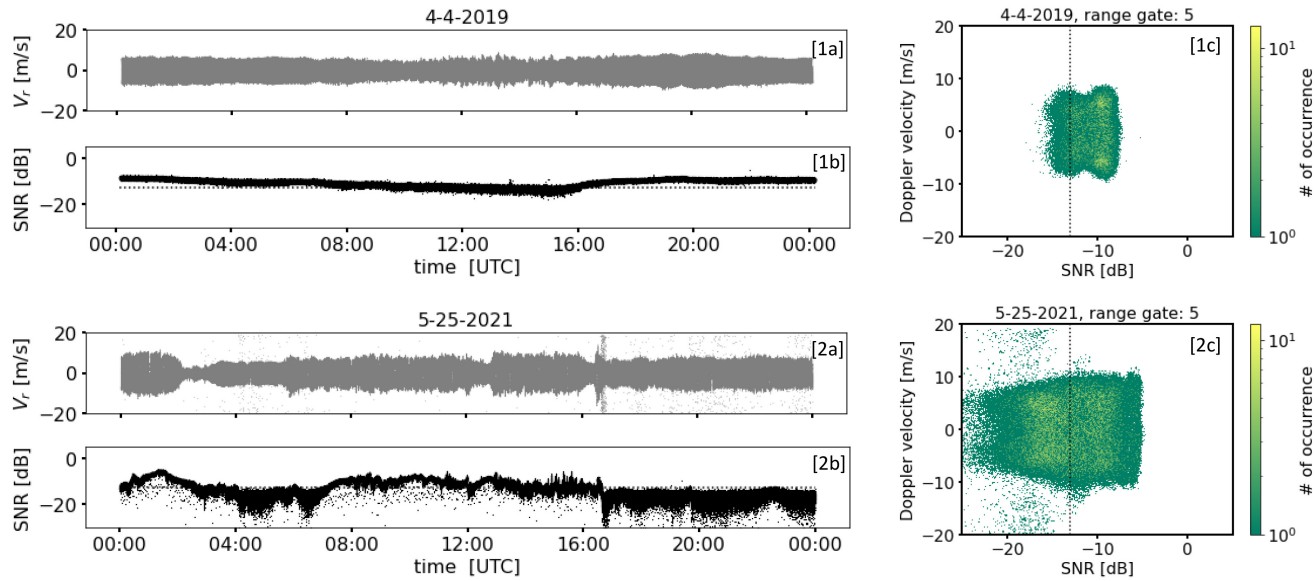

**Figure (7).** Time series plots of Doppler lidar radial velocity $V_r$ ([1a], [2a]) and corresponding SNR values ([1b], [2b]) from a conically scanning Doppler lidar for one day, i.e. 04-04-2019, with good measurement conditions which are reflected in relatively high SNR values and another day, i.e. 05-25-2021, with unfavorable measurement conditions which are reflected in relatively low SNR values. The corresponding $(V_r,$ SNR) plots are shown in [1c], [2c]. The dotted lines in [1b],[1c] and [2b],[2c] indicate the SNR threshold value $\mathrm{SNR}_{thresh}$ = - 12.7dB.



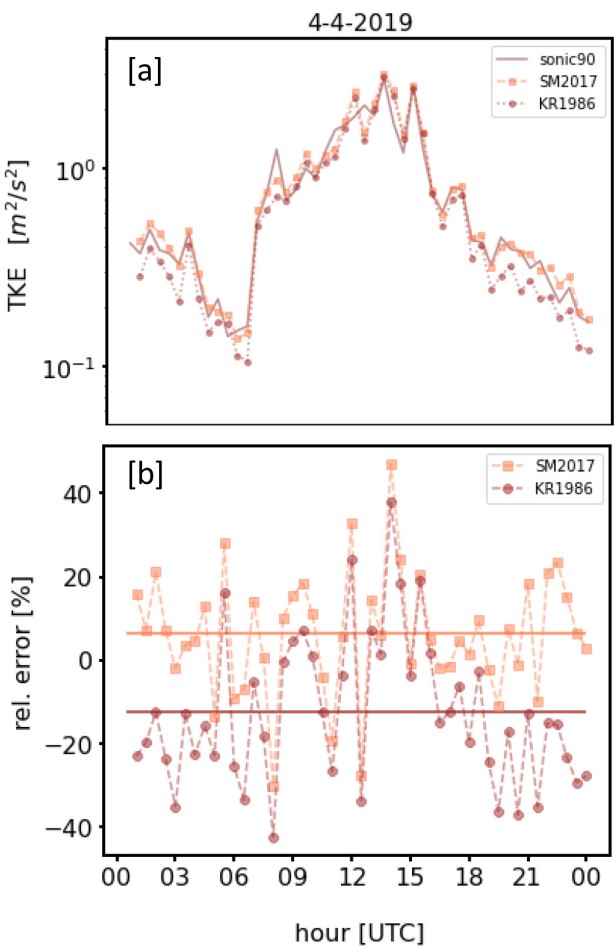

**Figure (8).** [a]: Time series of TKE at 90 m height derived from sonic anemometer measurements and from DL data using the methods proposed by Kropfli (1986) (KR1986) and Smalikho and Banakh (2017) (SM2017), respectively, for April 04, 2019. [b] Relative error between Doppler lidar TKE and sonic TKE for the sample day shown in [a]. The horizontal lines indicate the corresponding mean relative error over the whole day.

The method outlined in Smalikho and Banakh (2017) is building on a paper by Kropfli (1986) on the determination of TKE from radar measurements. A mere application of the Kropfli approach to DL based radial velocity data generally leads to an underestimation of the TKE mainly due to the finite laser pulse length effectively filtering small-sized eddies. Smalikho and Banakh (2017) introduced correction terms to account for this underestimation. The example day with reconstructed TKE values in comparison with independent sonic measurements at a measurement height of 90 m is shown in Fig. 8. To clarify

the effect of the corrections, TKE values without (KR1986 hereafter) and with the suggested corrections (SM2017 hereafter) are shown. With the KR1986 retrieval the TKE underestimation is most pronounced during night, i.e. while the atmospheric boundary layer is in stable state. Under stable conditions, smaller scale eddies primarily exist which obviously couldn't be fully





resolved over the pulse volume. During the day the corrections are smaller because the main fraction of the observed TKE was associated with larger and thus resolvable eddies. Although with SM2017 values of TKE are overestimated on average by
6.3%, the results are in better agreement with sonic data than the KR1986 values which underestimate the TKE by -12.45% on average. Therewith the finally corrected TKE is in good agreement with the sonic TKE which gives evidence for a proper functioning of the retrieval method itself. Note that the good agreement of the corrected TKE values with independent sonic measurements is also in accordance with the observations made by Smalikho and Banakh (2017) during a 5 day test-period with the particularity, that a forest fire was contributing to extremely favorable measurement conditions with high SNR in their
study.

Based on the above findings we can conclude that if differently filtered DL data are used as input for the retrieval process, large errors in the retrieved TKE have to be attributed to either a faulty noise filtering that leaves "bad" estimates in the filtered data set or to an overfiltering that removes too many reliable data points.

### 3.3   Effect of different filter methods on turbulence products

Depending on the used filtering techniques (see Sect. 3.1.1 and 3.1.2) the decision about which of the radial velocities are classified as "good" or "bad" estimates can turn out very differently. Hence, a comparison of retrieved turbulence products based on differently pre-filtered input data with independent reference measurements indirectly can yield insights into whether the filtering technique is useful to provide noise reduced data sets suitable for a retrieval of turbulence variables. To show this, the results of an inter-comparison of TKE variables based on differently pre-filtered DL data compared against sonic TKE
measurements are summarized in Fig. 9. Three different cases concerning the use of pre-filtered input data have been analyzed: (i) unfiltered input data (Fig. 9 [a]-[d]), (ii) SNR threshold based filtered input data using $\mathrm{SNR}_{thresh}$ = -12.7 dB (Fig. 9 [e]-[h]) and (iii) CNS based filtered data (Fig. 9 [i]-[l]). For each of the cases the retrieved TKE values are compared with tower based sonic anemometer measurements at 90 m height by means of different visualization types to give detailed insights into the quality and availability of the TKE products. Note that an additional distinction is made between TKE products which are
subject to different quality control (QC) steps. Here, the minimum requirement on a TKE value representing a 30 min mean is that its retrieval is based on more than 60 % of reliable measurements of $V_r$ (*lev_c* hereafter). Since the TKE reconstruction method relies on a variety of theoretical assumptions (e.g. see Eq. (22) in Smalikho and Banakh (2017)) a further QC step proves the fulfillment of these requirements (*lev_b* hereafter). To obtain meaningful results the evaluations are based on a two-month data set of DL measurements at the GM Falkenberg boundary layer field site which have been collected during
FESSTVaL from May 18 to July 17, 2021. In the following single subfigures of Fig. 9 are discussed in more detail.

The extent of grossly erroneous retrieved TKE values based on unfiltered "bad" estimates becomes obvious from the scatter plot example Fig. 9 [a] by considering *lev_c* data. These results demonstrate very clearly the general need of a careful pre-filtering of DL data prior to the retrieval of turbulence variables. Interestingly, with the additional QC step from *lev_c* towards *lev_b* the remaining data points are distributed much better along the 1:1 line. This underlines the importance and effectiveness
of further QC tests to distinguish between reliable and erroneous TKE values due to non-fulfillment of the retrieval assumptions. Notably less erroneous *lev_c* TKE values can be obtained if a pre-filtering is done either by the SNR thresholding or the CNS





**Figure (9).** Comparison of TKE from DL at 95 m height with data from a sonic at 90 m height over the period from 5-18-2021 to 7-17-2021. Each column represents the comparison of different TKE products based on differently pre-filtered input data. Unfiltered data have been used in the left column, SNR-threshold filtered data with $SNR_{thresh}$ = -12.7 dB and consensus (CNS) filtered data have been used in the middle and right columns, respectively. The subfigures of each column represent comparisons between DL and sonic data using different visualization techniques. These are from top to bottom: scatterplots, time series plots (one day (5-25-2021), only), Bland Altmann plots and histograms. The abbreviations "QC: lev_x" in the scatterplots ([a],[e],[i]) refer to different levels of product quality control (see Sect. 3.3).





technique even if there are differences w.r.t. data availability and quality. Comparing only *lev_b* TKE values with each other, best agreement with the sonic reference is obtained with $R^2 = 0.99$ and RMSD = 0.21 for pre-filtered DL data using the SNR thresholding technique (Fig. 9 [e]). TKE values based on CNS pre-filtered DL data have a slightly worse data quality with

$R^2 = 0.97$ and RMSD = 0.27 (Fig. 9 [i]) but score with an almost doubled data availability of 78.85 % ([l]) compared to the data availability of 36.77 % for the data set based on pre-filtered DL data using the SNR thresholding technique (Fig. 9 [h]). In order to discuss the reason for the reduced quality of TKE values based on CNS pre-filtered DL data, by way of example, the value pairs of the TKE comparisons for one single day (i.e. 05-25-2021) are highlighted in orange dots, additionally (see Fig. 9 [a], [e], [i]). The corresponding direct comparison of the time-series is shown in the subfigures [b], [f], [j] of Fig. 9.

It is important to point out that this day is a typical example for noise contaminated DL measurements under unfavourable atmospheric conditions which can be verified by means of the associated time series plots for the DL measurement variables $V_{meas}$ and SNR shown in Fig. 7. Here, the time series of $V_{meas}$ over the whole day shows the occurrence of "bad" estimates with a higher density in the early morning between 04:00 and 07:00 UTC and in the afternoon around 20:00 UTC. The corresponding SNR plots clearly show comparably low SNR values during these periods. Having this in mind and turning back

to the subfigures [b],[f],[j] of Fig. 9 the significance of a careful filtering of "bad" estimates in a pre-processing step becomes obvious again. If no filtering of "bad" estimates is performed, largest errors in the DL derived TKE values with the QC level *lev_c* occur during the periods with a high density of "bad" estimates (Fig. 9[b] and Fig. 7). Good agreement between DL derived TKE and sonic reference measurements is observed during the day, when almost no "bad" estimates appear in the measured data set. Note that gaps may be also be due to nonphysical negative retrieved TKE values which were set to NaN.

Using pre-filtered data based on the SNR thresholding technique (Fig. 9[f]) the retrieval approach generates a higher number of physically plausible TKE values during the day than obtained for the non-filtered input data set (Fig. 9[b]). For the noise contaminated time periods no derived TKE values are available because the SNR for most of the measured data within these periods was below $SNR_{thresh}$ = -12.7 dB (see Fig. 7). Using CNS pre-filtered data for the TKE retrieval the data availability is further enhanced so that there are hardly any data gaps throughout the whole day. Particularly for the noise contaminated time

period in the evening the availability and quality of the retrieved TKE values is very good. This is also an indirect indication that with the SNR thresholding technique a huge fraction of "good" estimates must have been rejected during this period. To verify this see also the VAD and SNR plots for the 30min measurement example on 05-25-2021 at 16:56 UTC in Fig. 1. Nevertheless, the overall quality of the TKE values based on CNS pre-filtered DL data is less compared to TKE values based on pre-filtered DL data using the SNR thresholding technique. The reason for this can be explained by means of the TKE

values obtained at 04:30 UTC and 16:00 UTC (see 9 [f],[j]) as an example. At these times the measured sonic TKE values are either very low ($\sim$ 0.015 m$^2$ s$^{-2}$) or very high ($\sim$ 7 m$^2$ s$^{-2}$) and the corresponding retrieved TKE values based on CNS pre-filtered DL data show either a pronounced over- or underestimation. Recalling, that the CNS filter performance of the corresponding 30-minute measurement intervals has already been analyzed in Sect. 3.1.2, this over- and underestimation can be easily explained. At 04:30 UTC the value $\Delta V_r = 3$ m s$^{-1}$ was too large so that "bad" estimates occurring within this range

remain in the data set and introduce an additional variance contribution yielding an overestimation of the TKE. At 16:00 UTC the value for $\Delta V_r$ was too small so that relevant features of the wind field were not captured. As a consequence the data set and





thus the derived variances are not representative for the 30min measurement interval yielding an underestimation of the TKE value. Hence, these examples clearly show the weaknesses of the CNS filtering technique if the $\Delta V_r$ threshold is inadequately selected. With a fixed $\Delta V_r$ during 24/7 routine measurements such situations occur quite frequently which explains why the data quality of TKE values retrieved from CNS pre-filtered DL data is worse compared to TKE retrieved from pre-filtered DL data using the SNR thresholding technique. One final comment should be given on the high sonic TKE value of $\sim 7 \, \mathrm{m^2 \, s^{-2}}$ at 16:00 UTC. This value is likely caused by an instationary wind field during the 30 min measuring interval rather than due to eddies in the turbulent flow. For that reason this value should be flagged as non-reliable. The problem, however, is that with the CNS filtering-technique relevant wind information characterizing this instationarity has been rejected so that the identification of such an instationarity wouldn't be possible.

Another possibility to assess the comparability between two measurement methods is a statistical method introduced by Bland and Altman (1986) which is frequently used in clinical laboratory to assess agreement between two methods of clinical measurements. The advantage of this method is that it not only provides insight into systematic deviations (bias) and limits of agreement but also reveals how the differences between two measurement methods depend on the magnitude of the measurement value. The plots [c], [g], [k] shown in Fig. 9 represent so called Bland-Altmann plots for a comparison of TKE products based on differently pre-filtered DL data with sonic measurements. Here, the y-axis shows the percentage error (PE) between two paired measurements, i.e. 100*(TKE$_{DL}$ - TKE$_{Sonic}$)/TKE$_{Sonic}$, and the x-axis represents the average of these values, i.e. (TKE$_{DL}$+TKE$_{Sonic}$)/2. Additionally, the horizontal lines indicate the mean percentage error ($\overline{PE}$) and twice the standard deviation ($\pm 2\sigma$) from the mean as statistical limits of agreement. For the Bland-Altmann plot showing a comparison of the TKE values retrieved from SNR pre-filtered DL data with sonic measurements, the mean percentage error is close to zero ($\overline{PE}$ = -0.84%) (Fig. 9 [g]). This indicates that there are hardly any systematic differences between the two measurements. Additionally it can be seen that there is a nearly symmetric distribution of the paired points around the horizonatl line indicating $\overline{PE}$ with a slight tendency towards a more frequent underestimation by the DL for small TKE values and a more frequent overestimation for higher TKE values. As a result, in 95% of the cases the DL delivers a TKE value that is up to 39% smaller or up to 37 % larger than the one measured by the sonic. One gets slightly different results ($\overline{PE}$ = - 0.63 %, $\overline{PE} \pm 2\sigma$ = {+40 %, -41 %}) by considering the Bland-Altmann plot showing a comparison of TKE values retrieved from CNS pre-filtered data with sonic measurements (Fig. 9 [k]). In addition a slightly different point cloud distribution is noticeable. This somewhat poorer comparative statistic of TKE based on CNS-prefiltered data compared to TKE based on SNR-prefiltered data already reflects, to some extent, the shortcomings of CNS filtering discussed above, i.e., an underestimation of TKE in strong turbulence and an overestimation of TKE in weak turbulence in the case of a poorly chosen $\Delta V_r$ threshold. In the following, we show that these findings are further strengthened when TKE data from higher altitudes are additionally considered.

With increasing measurement heights the probability of "bad" estimates in the radial velocity measurements increases. This is due to typically decreasing aerosol density which is also noticeable in weaker SNR values. Hence the actual robustness of the filtering methods can be tested at larger measuring heights, where the methods have to cope with an increased occurrence of noise. Unfortunately, the tower based measurements at the GM Falkenberg provide measurements up to 99m height, only, so that for higher altitudes no independent references for a comparison are available. For that reason we were seeking for an

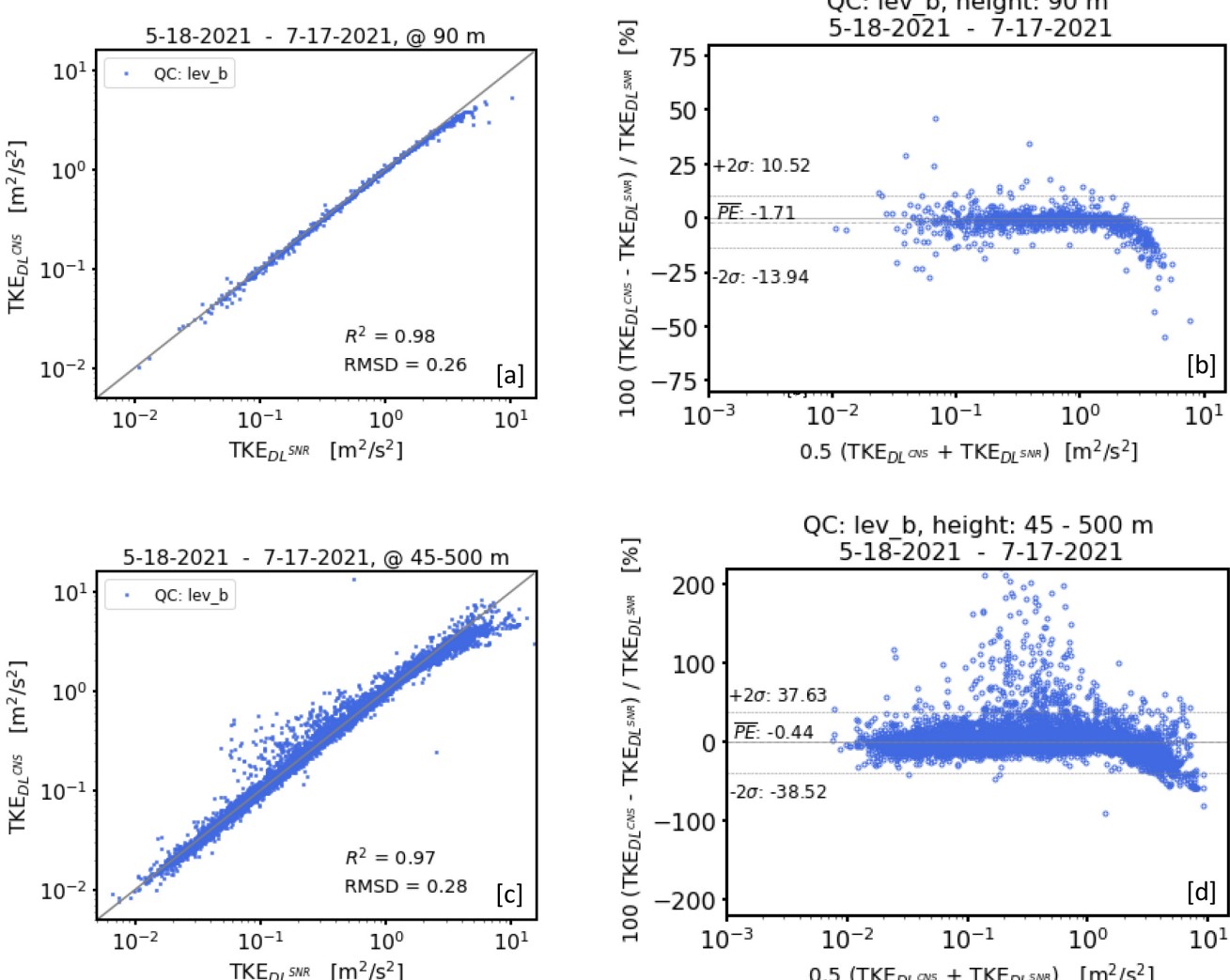

**Figure (10).** Comparison of TKE from DL based on differently pre-filtered measurement data prior to the TKE retrieval as proposed in Smalikho and Banakh (2017). The Scatterplots [a] and [c] show TKE products based on CNS pre-filtering ($TKE_{DL^{CNS}}$) against TKE products based on a pre-filtering using the SNR-thresholding technique ($TKE_{DL^{SNR}}$) with $SNR_{thresh}$ = -12.7 dB whereby [a] represents DL measurements at 95 m and [c] represents all available DL measurements up to 500 m height over a measurement period from 5-18-2021 to 7-17-2021. As for height availability same applies for the Bland and Altmann (BA) plots [b] and [d]. Here, the relative percentage difference between $TKE_{DL^{CNS}}$ and $TKE_{DL^{SNR}}$ products is plotted against its mean value. For more details concerning BA plots see Sect. 3.3

alternative way and make use of the following strategy. In the subsequent analysis we consider DL TKE values based on SNR threshold based filtered input data as an alternative reference intended to replace the missing sonic data at higher altitudes. This can be motivated by two arguments. On the one hand, it could be shown that the comparison of TKE values based on





SNR pre-filtering with independent sonic data provided the best results (see Fig. 9). On the other hand the SNR thresholding approach is a standard method to exclude "bad" estimates from DL measurements. For that reason confidence in the validity of TKE products based on SNR pre-filtering is very high and TKE products based on differently pre-filtered input data should be comparable by similar performance of the filtering method. A comparison of TKE products based on SNR pre-filtered measurement data with those based on CNS filtering at 95m height and for all measurement heights between 45m and 500

m is shown in Fig. 10 [a],[b] and ([c],[d]), respectively. In Fig. 10 [a],[b] generally less scatter of the data points compared to the scatterplots shown in Fig. 9 [e],[g] and [i],[k] is striking. This can be explained by the fact that regardless of the type of filtering, in Fig. 10 measurements from the same measuring system based on the same physical measuring principle are compared. In contrast, in Fig. 9 a comparison of measurements taken from different measuring systems (i.e. sonic vs. DL) with different underlying measuring principles (i.e. point vs. volume measurement) is shown. In this respect, the comparison

of TKE values based on different pre-filtering methods but the same measuring system is even better suited to work out the differences between the filter methods. A fundamental problem that becomes apparent in 10 [a],[b] is the clear underestimation of larger TKE values based on CNS pre-filtering. Note that there are also indications of this in Fig. 9. The level up from which the underestimation starts to become significant is at TKE values above 2 $\mathrm{m}^2\,\mathrm{s}^{-2}$. This finding is reinforced by considering the subfigures 10 [c],[d] which include TKE values from all heights up to 500m. Here, an increased overestimation of TKE values

in a range between 0.06 $\mathrm{m}^2\,\mathrm{s}^{-2}$ < TKE < 0.5 $\mathrm{m}^2\,\mathrm{s}^{-2}$ can be found, additionally. An explanation for this has already been given above and in Sect. 3.1.2.

## 4 Ideas for new filtering techniques

The drawbacks of the filter techniques discussed in the previous section motivate our work to seek new filter methods. These should realize an efficient noise filtering and equally provide the highest possible data availability. In this section we will

present two different approaches for new filter techniques which have to be applied depending on the measurement's noise characteristics. The filtering techniques are referred to as **approach I** and **approach II**, hereafter. An overview illustrating the basic structure of the two filtering techniques is given in Fig. 11. Each approach consists of two parts, namely a first **coarse filter** $(\mathbf{I+II})$ yielding first guess results for "good" estimates and a second **filter for post processing**. Latter builds on the first guess results and is utilized for further optimization. For each individual filter step a different perspective

of data representation is used to analyze the occurrence of "bad" estimates in the DL measurements. The coarse filter works using the newly introduced framework of the VV90D perspective. More details on this framework will be given in Sect. 4.1. The application of the post-processing filter requires the VAD perspective.

### 4.1 Framework of the VV90D perspective

Radial velocity measurements from a conically scanning DL plotted in a $V_r - t$ diagram typically show a sinusoidal graph (Figs.

1 [a], [e], [i]). In classical mechanics, the harmonic oscillator is a well known physical system showing similar behaviour, i.e., with evolving time the motion and displacement of the oscillating system also matches the pattern of a sine curve. For the





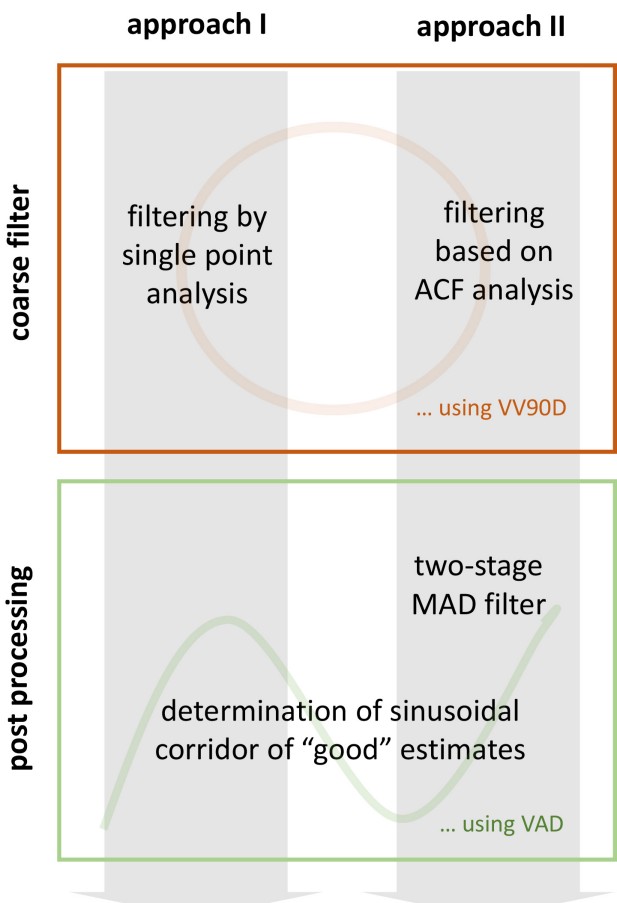

**Figure (11).** Schematic overview illustrating both the two part structure of the two filtering techniques **approach I** and **approach II** and the key ideas of the individual filter approaches.

combined view on the time evolution of both variables a phase-space diagram is frequently used. Due to a 90° phase relation-ship between motion and displacement, the phase-space diagram for a harmonic motion typically shows circular trajectories (Vogel, 1997). The concept of data analysis introduced in this section builds on the idea of representing the DL radial velocity

measurements in analogy to this phase-space diagram perspective. In particular, a so-called VV90D diagram is introduced. In this diagram the time series of radial velocity measurements $V = V_r(R, \theta; t)$ obtained from a conically scanning DL is plotted along the y-axis and the same measurement series but with a phase shift by 90°, i.e. V90$= V_r(R, \theta - 90°; t)$, is plotted along the x-axis. Compared to the commonly used VAD visualization technique for DL radial velocity measurements, the VV90D represents an alternative way of displaying radial velocity measurements from a conically scanning DL. It will be shown that

this offers a new perspective on the measurement data and opens up new possibilities for data analysis at the same time.





Measurement examples from a conically scanning DL displayed using the framework of the VV90D perspective are shown in Fig. 12. In order to clarify the different view on the measurement data first, the examples shown here are limited to noise-free cases which differ from each other in terms of its wind properties. While the plots in Fig. 12 [a]-[c] illustrate a homogeneous and stationary case (case 1), the plots in Fig. 12 [d]-[f] represent a non-stationary measurement example (case 2). For each

case the classical $V_r - t$ diagram is shown first (see Fig. 12 [a], [d], respectively). The typical picture is a smooth sinusoidal course of the radial velocity $V_r$ with a nearly constant amplitude for case 1 and a sinusoidal course superimposed with smaller fluctuations and a varying amplitude for case 2. Same measurements visualized by means of the VV90D diagram (see Fig. 12 [b],[e], respectively) show circular structures as a typical pattern generated by the paired data points (x=V90,y=V). For case 1 the graph reflects a relatively thin circular ring and for case 2 the graph shows a slightly wider and more blurred ring structure.

A more quantitative description of the above described ring structures in the VV90D is given with Fig. 12 [c] and [f], respectively. The diagrams shown here are referred as $r_i$ - $count\{V_i\}$ diagram, hereafter. Here, $r_i$ denotes the radius of a pre-defined circular ring $r_i - \Delta r \leq r_i \leq r_i + \Delta r$ with origin at (x=0,y=0) and width $\Delta r$ in the VV90 plane. The quantity $count\{V_i\}$ denotes the number of measurement data that can be found in this ring. Note that these data represent a circular ring related subset $V_i$ of the whole measurement series, i.e. $V_i \subset \{V\}$. Taking the equation of circle into account, i.e. $r = x^2 + y^2$, in practice

both the subset $V_i$ and $count\{V_i\}$ can be determined by identifying those radial velocities $V = V_r(R, \theta; t)$ of the measurement time series that satisfy the relation

$$r = \sqrt{V90^2 + V^2} \tag{7}$$

with the range of radii $r$ defined through the boundaries of the circular ring $r_i - \Delta r \leq r_i \leq r_i + \Delta r$. In order to generate the $r_i$ - $count\{V_i\}$ diagrams shown in Fig. 12 ([c],[f]) the full area of the VV90 plane has been subdivided into closely spaced

circular rings of increasing radius ($i = 0, ..., n$ with $n \in \mathbb{N}$) with discrete fixed steps $\Delta r = 0.5$ m s$^{-1}$. For case 1, the data availability of measured radial velocities is constraint to only a few circular rings with $r_i$ ranging between 5 m s$^{-1}$ and 6.5 m s$^{-1}$ with the largest fraction of measurements in the circular ring with $r_i = 5.5$ m s$^{-1}$ (see Fig. 12 [c]). Additionally, due to the stationary wind field conditions the obtained availability distribution is strictly uni-modal and symmetric. For case 2 the measurements are distributed over a broader range of circular rings with $r_i$ taking values between 0.5 m s$^{-1}$ and 12.5 m s$^{-1}$.

The largest fraction of the measurements occurs within a circular ring with $r_i = 4.5$ m s$^{-1}$ (see Fig. 12 [f]). The distribution of data availability is nearly uni-modal but asymmetric.

In the following we refer to the VV90D diagram and the corresponding quantitative analysis of the included measurement points by means of the $r_i$ - $count\{V_i\}$ diagram as the framework of the VV90D perspective. It will be shown that this framework helps to achieve a suitable filtering of DL radial velocities also under weak signal conditions. The examples shown in

Fig. 12 represent just two specific situations, and a great variety of VV90D and corresponding $r_i$ - $count\{V_i\}$ diagrams may result for different atmospheric and lidar signal conditions, including multi-modal distributions (not shown here).





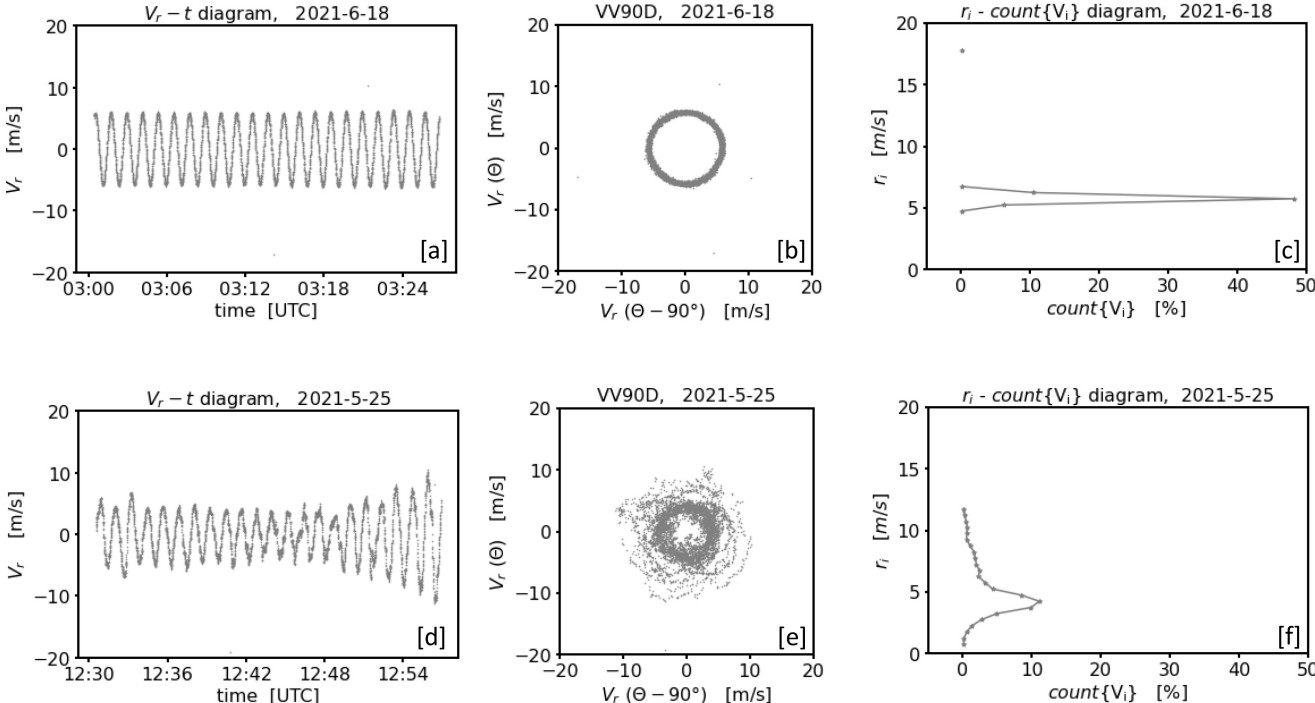

**Figure (12).** Examples for a graphical visualization of radial velocity measurements from a conically scanning DL using the framework of the VV90D perspective for two cases with stationary (upper row) and non-stationary (lower row) winds, respectively. The sub-figures in each row illustrate in sequence: a time series plot of the measured radial velocity over a 30 min time period ([a],[d]), same data plotted using the VV90D perspective ([b],[e]) and the frequency distribution of data points in the VV90 plane binned by circular rings $r_i - \Delta r \leq r_i \leq r_i + \Delta r$ where $i = 1,..,n$ ( $n \in \mathbb{N}$) and $\Delta r = 0.5$ m s$^{-1}$.

## 4.2 Coarse filtering techniques

In this section we provide a detailed overview of the first parts of the two filter approaches illustrated in Fig. 11. In order to work out that data sets with different "bad" estimate distributions require different filtering techniques this overview will be

given based on the noise contaminated DL measurement examples of type A and type B introduced in Sect. 2.2. Note that each approach makes use of the framework of the VV90D perspective.

### 4.2.1 Filtering by single point analysis - coarse filter I

For an introduction of the first coarse filter, examples of noise contaminated DL measurement using the framework of the VV90D perspective are shown in Fig. 13. Compared to the the noise-free measurement examples (see Fig. 12) the most

noticeable feature in the VV90D is the greater spread of paired DL data (x=V90,y=V) (see Fig. 13 [a] and [d], respectively) which can be explained by the occurrence of noise. Furthermore, the characteristic ring structures as discussed for the noise-free measurement examples appear less pronounced and are rather diffuse if noise is included. Accordingly, the corresponding





$r_i$ - $count\{\mathsf{V}_i\}$ diagrams (see Fig. 13 [b] and [e], respectively) show a somewhat broader distribution of data points with more gently sloping flanks over a larger range of circular rings than compared to the noise-free cases (see Fig. 12 [c],[f]). Eventually

it should be pointed out that for the type B noise measurement example the more densely distributed data points around zero make a cross-shaped region visible in the VV90D (Fig. 13 [d]) and cause a pronounced secondary peak in the $r_i$ - $count\{\mathsf{V}_i\}$ diagram (Fig. 13 [e]).

In order to motivate the first coarse filtering technique, the classical time series plot is also shown (see Fig. 13 [c] and [f], respectively) with an additional labeling of data points occurring in three differently selected circular rings. Based on

this labeling it becomes obvious that circular rings with high data availability contain a huge fraction of reliable DL radial velocities, i.e. "good" estimates, and contain almost no "bad" estimates. The data occurring in these rings follow the expected sinusoidal course of the DL measurements in a dense sequence of points (see the black dots in Fig. 13 [c],[f]). For circular rings with increasingly lower data availability, however, the associated subsets $\mathsf{V}_i$ contain increasingly more "bad" estimates which can be seen from the fact that some data points of these subsets occasionally deviate more from the sinusoidal course (see

orange dots in Fig. 13 [c],[f]). Note that these findings apply mostly to the type A noise measurement example. Considering the measurement example contaminated with type B noise a high fraction of "bad" estimates generated by an increased occurrence of "noise around zero" causes an additional secondary peak in the $r_i$ - $count\{\mathsf{V}_i\}$ diagram, clearly separated from the peak representing the real wind (see the red marks in Fig. 13 [e],[f]). Except for the latter from the analysis above we may conclude: If the subsets $\mathsf{V}_i$ of data points binned by circular rings $r_i - \Delta r \leq r_i \leq r_i + \Delta r$ are analysed individually, "good" estimates

mostly occur in a dense sequence of points following the sinusoidal course of the measurements. In contrast, "bad" estimates mostly occur as singular points having no further data points in its immediate environment while taking any value within the velocity space $\pm\,20\,\mathrm{m\,s^{-1}}$.

The properties described above open a first way for the development of a filter technique for "bad" estimates, namely, by detecting and discarding singular points in circular-ring-related subsets $\mathsf{V}_i$ of measurements. Practically, this can be implemented

as follows: Use the framework of the VV90D perspective. Consider all circular rings spanning the VV90 plane individually. To select the ring-related data points, always start with the original time series and set all those measurement points to a non-numeric flag value (e.g. NaN) which do not satisfy Eq. (7). This gives for each circular ring a certain ring-specific time series which has the length of the original one but where only those measurement points are allocated with a numerical value, which occur in the respective circular ring. Then, for each of the ring-specific time series check sequentially each position of the time

series for flagged predecessor and successor positions within a pre-defined azimuthal environment. Positions occupied by an unflagged value, i.e. a numerical value, but with flagged predecessor and successor positions can be regarded as a singular point and discarded. Finally, the resulting circular-ring-related time series have to be merged back to one full time series which then represents a filtered time series where most of the "bad" estimates should be excluded. This filtering technique is referred to as **coarse filter I**, hereafter. Note, however, that not all "bad" estimates necessarily occur as singular points. Hence, it

is possible that a minor portion of "bad" estimates will still remain in the measurement series. Those can be discarded using classical outlier detection methods (e.g. the $3\sigma$ rule applied to differences of radial velocity measurements of two consecutive azimuthal measurement points) which are only effective if outlier are real outlier in the sense that they represent only a few



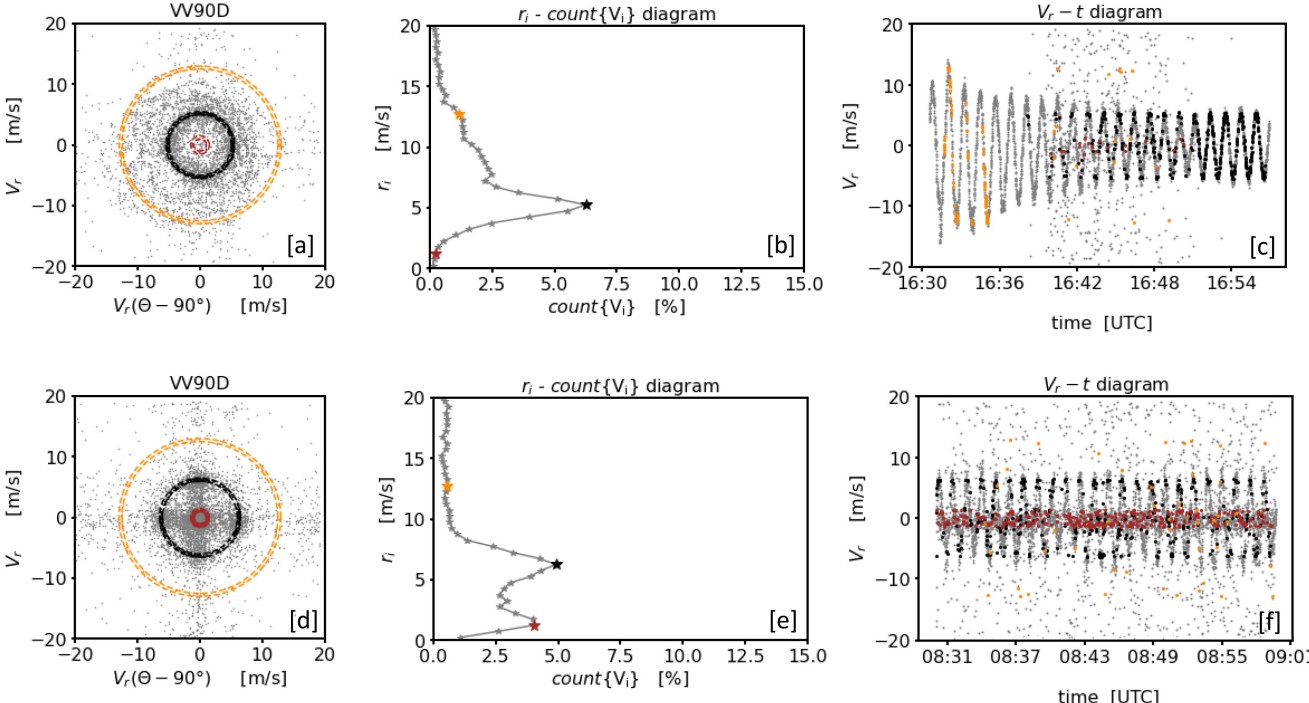

**Figure (13).** Examples of noise contaminated DL measurements over a 30 min time period analysed using the framework of the VV90D perspective. The upper and lower rows show measurements contaminated with type A and type B noise, respectively (see Fig. 1). The plots in each row show from left to right: the VV90D plot ([a],[d]), the frequency distribution binned by circular rings $r_i - \Delta r \leq r_i \leq r_i + \Delta r$ where $i = 1,..,n$ ( $n \in \mathbb{N}$ ) and with $\Delta r = 0.5$ m s$^{-1}$ ([b],[e]) and the time series plot ([c],[f]). Additionally for each measurement example three specific circular rings have been chosen to illustrate where the measurement data contained in the circular ring (highlighted in different colors) are located in time series plot.

unusual observations. In the case of noise-contaminated measurements the fraction of "bad" estimates is too high, which would not justify considering them as outliers in the original sense.

Results that can be obtained using **coarse filter I** applied to the measurement examples of Fig. 13 are shown in Fig. 14. It turns out that "bad" estimates can be best removed from measurements including type A noise (Fig. 14[a]). During the sub-interval of enhanced noise in the time series of the radial velocities, however, a severe thinning of data is striking, which reflect "good" estimates (Fig. 14 [b]). This erroneous exclusion of "good" estimates happens when the availability of data, i.e. $count\{V_i\}$, is generally comparatively low for a larger number of circular rings. That is because low values of $count\{V_i\}$ also

mean that there is an increased probability that "good" estimates appear more frequently as singular points. Unfortunately, the latter makes the performance of **coarse filter I** the weaker the more "bad" estimates are included in the time series of Doppler velocities (Fig. 14 [d], [e]) since an increased number of "bad" estimates increases the number of sparsely filled circular rings extending over the whole VV90 plane. From the VAD perspective, however, it can be seen that despite the strong data thinning the remaining data points represent a suitable first guess for "good" estimates of the time series which reveal the range of



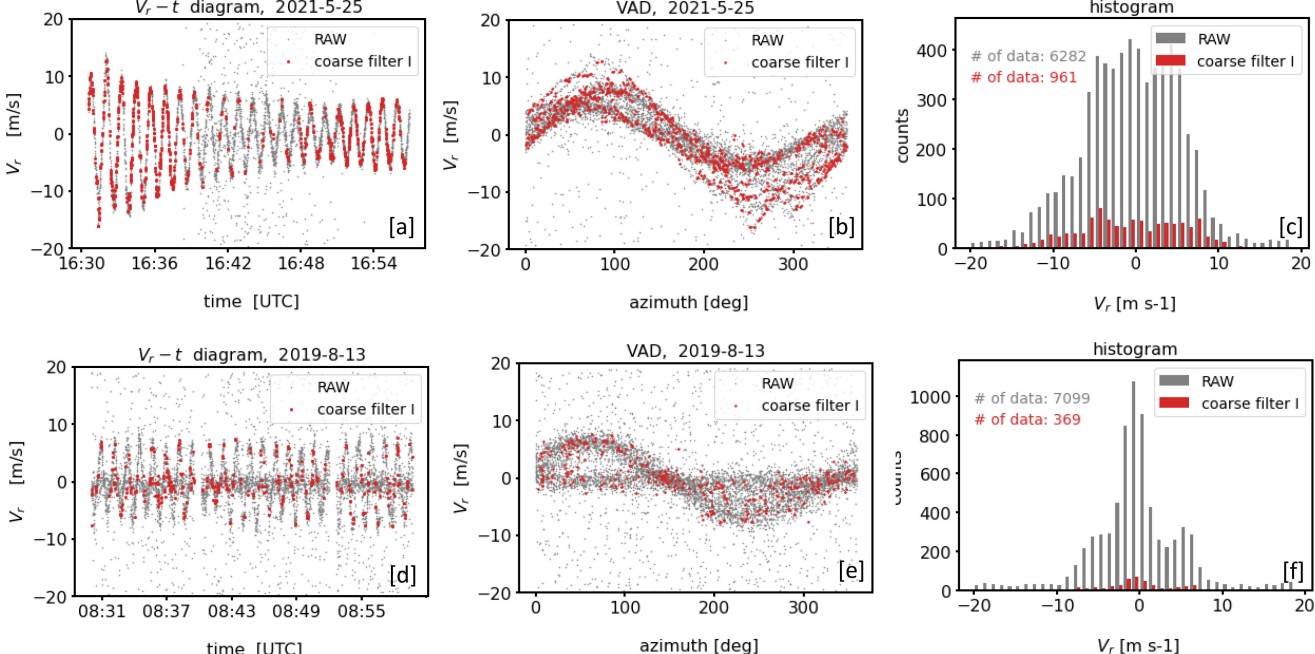

**Figure (14).** Examples for the outcome of **coarse filter I**. The upper and lower rows show measurements contaminated with type A and type B noise, respectively (see Fig. 1). The plots in each row show from left to right: a comparison of the time series before (RAW) and after the application of **coarse filter I** using a $V_r - t$ diagram ([a],[d]), a comparison of the time series before and after filtering using the VAD perspective ([b],[e]) and a comparison of the associated histograms of the radial velocities ([c],[f])

"good" estimates for each azimuthal direction (Fig. 14 [b]). Considering the results of **coarse filter I** for the type B noise example the performance of the filtering technique is not convincing. Here, a huge fraction of "bad" estimates belonging to the type of "noise around zero" remains in the data set after applying the filtering method (Fig. 14 [e]). From that we conclude that the specific distribution characteristics of type B noise measurements prohibits the possibility to distinguish between "good" and "bad" estimates by means of singular point detection.

### 4.2.2 Filtering based on ACF analysis - coarse filter II

In Sect. 2.2.2 it has been shown that the ACF calculated from the measurement time series of a conically scanning DL can give useful indications about the occurrence of noise in general, but without precise knowledge about which of the measurements are "bad" estimates. It will be shown next, that using the framework of the VV90D perspective (see Sect. 4.1) the particular noise information provided by the ACF for circular ring specific subsets of the measurement series can be used to make a better

localization of "bad" estimates possible.

In the foregoing sections it has already been shown that data points belonging to subsets of the time series of DL measurements at circular rings $r_i - \Delta r \leq r_i \leq r_i + \Delta r$ with comparatively high data availability $count\{V_i\}$ generally describe



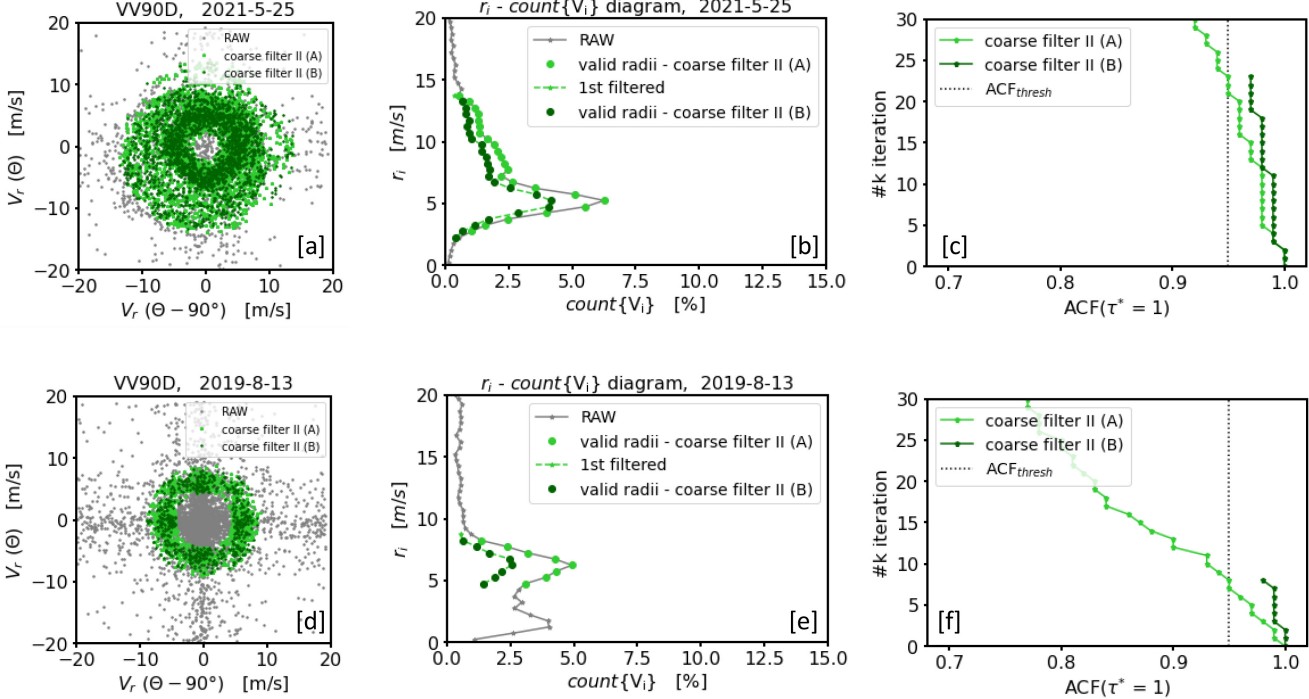

**Figure (15).** Example for the treatment of noise contaminated DL measurements over a 30 min time period illustrated using the framework of the VV90D perspective in combination with intermediate results when **coarse filter II** has been applied. The upper and lower rows show measurements contaminated with type A and type B noise, respectively (see Fig. 1). The plots in each row show from left to right: the VV90D plot ([a],[d]), the frequency distribution binned by circular rings $r_i - \Delta r \leq r_i \leq r_i + \Delta r$ where $i = 1,..,n$ ( $n \in \mathbb{N}$) and with $\Delta r = 0.5\,\mathrm{m\,s^{-1}}$ ([b],[e]) and a graphic illustrating the change of the ACF with each iteration step (for more details see Sect. (4.2.2)). Additionally, intermediate results from two consecutive applications of **coarse filter II**, indicated by A and B, are shown.

a sinusoidal course in the $V_r - t$ diagram (see Sect. 2.2.2) and thus mostly reflect "good" estimates. Accordingly, it is to be expected that an evaluation of the ACF of these subsets at the first lag index would yield $\mathrm{ACF}(\tau = 1; i) \sim 1$ (see Sect. 2.2.2).

A possible filter approach could therefore be to locate those circular rings where the ACF of the time series of the associated measurement data does not fall below a pre-defined threshold ($\mathrm{ACF}_{thresh}$ hereafter). Practically, this can be implemented as follows: Use the framework of the VV90D perspective. Seek in a first step for that circular ring $r_i - \Delta r \leq r_i \leq r_i + \Delta r$ in the $r_i$ - $count\{V_i\}$ diagram with an absolute maximum in the data availability (i.e. seek for $r_i$ with $MAX(count\{V_i\})$. Next, set temporarily all data points of the original measurement time series $\mathrm{V} = V_r(R, \theta; t)$ to a non-numeric flag value (e.g. NaN)

which do not satisfy Eq. (7) for the circular ring with the central radius $r_i$ previously determined. This gives an initial guess for a filtered time series $\mathrm{V}^f = V_r^f(R, \theta; t)$. To proof a low noise contamination by means of the ACF, replace the flagged positions of the time series $V^f$ by an estimated numerical value from the respective unflagged predecessor and successor positions using linear interpolation and calculate the ACF. The replacement of flagged positions is a necessary technical step to maintain the length of the time series and therewith the azimuthal distances between the single measurement points of the series. Latter is





important since consecutive measurement data with different azimuthal distances would correlate differently with each other, which in turn would affect the ACF (see Sect. 2.2.2). If the good quality of this filtered time series is reasonably assured, i.e. if $ACF^f(\tau = 1) \sim 1$, the unflagged values of $V^f = V_r^f(R,\theta;t)$ can be regarded as reliable. In the same way as just described, by means of further iteration steps it is possible to gradually increase the number of reliable measurement data by repeating the above described procedure taking not only the data from subsets $V_i$ at circular rings $r_i - \Delta r \leq r_i \leq r_i + \Delta r$ with

$MAX(count\{V_i\})$ into account but also those from adjacent circular rings $r_{i\pm1} - \Delta r \leq r_{i\pm1} \leq r_{i\pm1} + \Delta r$ whereas the data availability determines the order. This effectively results in the consideration of a wider circular ring with an accordingly higher number of data. Latter are constituents of a newly filtered time series $V^{f_{new}} = V_r^{f_{new}}(R,\theta;t)$ after the k'th iteration. As long as the added subsets $V_{i\pm1}$ from adjacent circular rings include mostly "good" estimates, the associated ACF of the newly generated times series will remain close to one, i.e. $ACF^{f_{new}}(\tau = 1) \sim 1$, and the iteration can be continued. The iteration has to be

stopped if the ACF of the newly generated time series falls below a pre-defined threshold, i.e. if $ACF^{f_{new}}(\tau = 1) < ACF_{thresh}$. That happens when the recently added data represent subsets of circular rings with an increased fraction of "bad" estimates. In this case, the result from the previous iteration step can be considered as the best possible filtered time series with a maximum possible data availability and a low proportion of bad estimates at the same time. The above proposed filtering technique is referred to as **coarse filter II**, hereafter. Further information that could be useful for practical implementation of **coarse**

**filter II** is given in Appendix D.

For the measurement examples shown in Fig. 13 relevant technical details concerning **coarse filter II** are shown in Fig. 15. The change of $ACF^{f_{new}}$ with each iteration step is illustrated in Fig. 15 [c] and [f]. The circular rings included with each iteration step until the iteration has been stopped and the associated data points of the subsets $V_i$ are colour-coded in the subfigures [b],[e] and [a],[d], respectively. The final filter results of the measurement series that can be obtained using

$ACF_{thresh} = 0.95$ are shown in Fig. 16. The filter results based on **coarse filter II** differ from the results obtained based on **coarse filter I** (see Fig. 14) in two aspects. On the one hand the erroneous rejection of "good" estimates is less than in comparison to the **coarse filter I**. This can be verified based on the histograms comparing both the distributions of the measurements from the original and the filtered time series which are shown in Fig. 14 [c], [f] and Fig. 16 [c], [f], respectively. Regarding the need of sufficient data availability required for a representative product retrieval this can be seen as an important

advantage of **coarse filter II**. On the other hand, however, it has to be noticed that a certain number of "bad" estimates could not be removed. This disadvantage of **coarse filter II** is more frequently observed for type A noise contaminated measurements if the conditions during the measurement interval are non-stationary (see Fig. 16 [a]-[b]) than for stationary intervals (not shown here). In this case an increase of the threshold value (e.g. $ACF_{thresh} = 0.99$) would help to better remove "bad" estimates, however, this would be at the expense of removing more "good" estimates which describe the non-stationarity

in the wind field. For type B noise contaminated measurement intervals this issue also occurs for non-stationary measurement intervals (see Fig. 16 [d]-[e]).





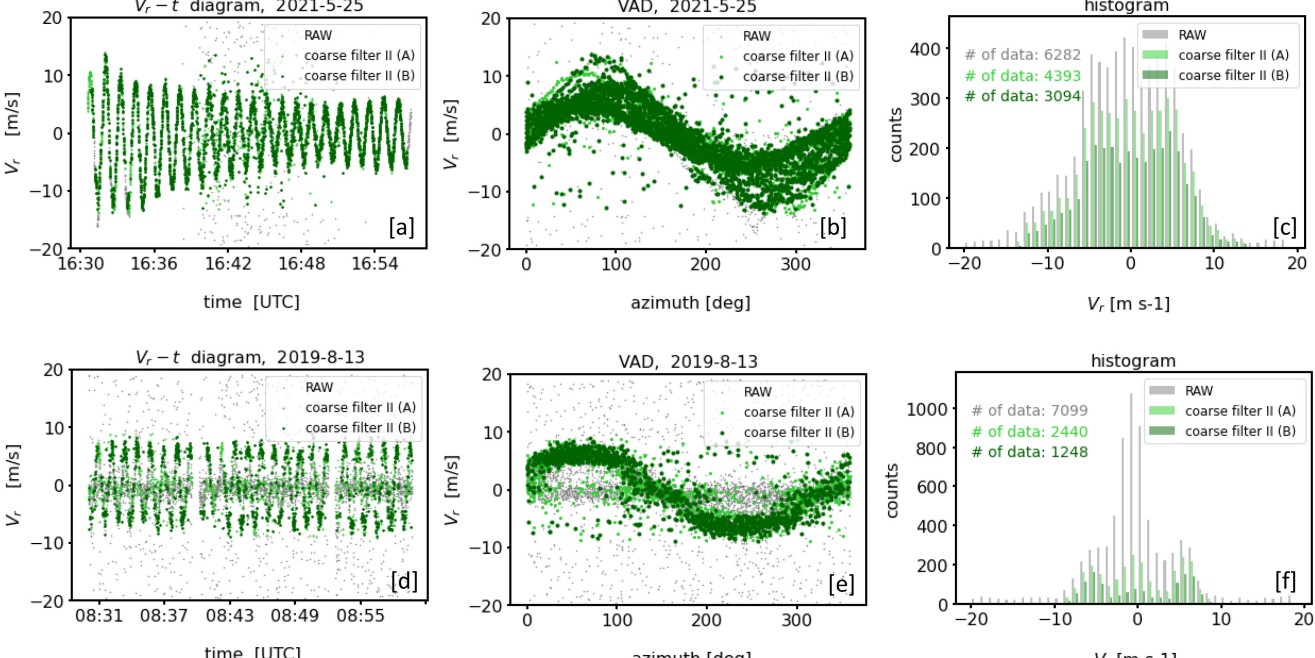

**Figure (16).** Examples for the outcome of **coarse filter II**. The upper and lower rows show measurements contaminated with type A and type B noise, respectively (see Fig. 1). The plots in each row show from left to right: a comparison of the time series before (RAW) and after application of **coarse filter II** ([a],[d]), a comparison of the time series before and after filtering using the VAD perspective ([b],[e]), and a comparison of the associated histograms of the radial velocities ([c],[f]).

## 4.3 Post processing filter for optimization

Two approaches for a coarse filtering technique have been introduced in Sect. 4.2. It has been shown that both offer quite some potential for the detection and elimination of "bad" radial velocity estimates. Nonetheless, the coarse filter results are

not yet satisfactory for the following reasons: Firstly, the frequently unjustified rejection of "good" estimates after apply-ing **coarse filter I** results in an unnecessary reduction of reliable measurement data while a high data availability is re-quired for a meaningful calculation of turbulence variables. Secondly, the number of remaining "bad" estimates after applying **coarse filter II** is still too high so that large errors in the retrieved turbulence variables can occur. Hence, additional efforts are required to further optimize the filter results. At this point, therefore, the results of **coarse filter I** and **II** will be treated

as intermediate results only. Possible further optimization steps are considered in more detail in this subsection. The entirety of these steps represent the **filter for post processing**. Note that all analyses are from now on carried out using the VAD perspective.





### 4.3.1 Two-stage MAD filter

The median absolute deviation ($MAD$) is a frequently used statistical tool to find outliers from measured data sets having
an unimodal and symmetrical distribution (see Sect. 3.1). Here the term outlier refers to a few uncontrollable and abnormal
observations which seem to lie outside the considered population. If $X = \{x_1, x_2, ..., x_n\}$ with n $\in \mathbb{N}$ denotes a given data set
of measurements the $MAD$ is defined through

$$MAD = median(\mid x_i - median(X) \mid) \qquad . \tag{8}$$

Outliers in X are detected by spanning an interval over the median plus/minus a cutoff value $q$ times $MAD$. Values $x_i$ which
are not included in the interval, i.e. which do not satisfy the relation

$$median(X) - q\,\frac{MAD}{0.6745} \; \leq \; x_i \; \leq median(X) \, + q\,\frac{MAD}{0.6745} \tag{9}$$

are regarded as outliers (Iglewicz and Hoaglin, 1993). The cutoff value q is mostly chosen arbitrarily. Iglewicz and Hoaglin
(1993) suggest $q = 3.5$. A modification of the $MAD$ is the so called *double MAD*, which can be used for non-symmetric
distributions (Rosenmai, 2013). The $MAD$ outlier detection method works in analogy to the $3\sigma$-rule of thumb (Gränicher,
1996) but is classified as the more robust one against outliers in the literature (Iglewicz and Hoaglin, 1993). Robust in this
context means that the $median$ and $MAD$ itself are less affected by outliers than the $mean$ or the standard deviation $\sigma$ which
is a statistic that measures the dispersion of a dataset relative to its $mean$.

Care has to be taken when applying the $MAD$ method to noise contaminated DL data. With an increasing fraction of "bad"
estimates for $V_r$ they can no longer be considered as only a few unusual observations so that the requirements for an application
of the $MAD$ are no longer met. Fortunately, in Sect. 4.2.2 it has been shown that the application of **coarse filter II** proved
successful to effectively reduce the fraction of "bad" estimates (see Fig. 16 [b], [e]). Therewith, the smaller number of remaining
"bad" estimates rather reflect what is meant an outlier so that an application of the $MAD$ filter technique as a follow-up filter
step of **coarse filter II** seems to be more justified than using the $MAD$ filter technique alone.

For a follow-up implementation of the $MAD$ filter technique we switch back from the VV90D perspective to the VAD
perspective and pursue an application of the $MAD$ outlier detection method in two stages. In the first stage (MAD_part_I,
hereafter) we apply the $MAD$ azimuthwise, i.e. to data sets representing measurements from only one direction, respectively.
In the second stage (MAD_part_II, hereafter) we apply the $MAD$ to a data set representing squared deviations $(V_r - V_r^{FSWF})^2$
of radial velocity measurements $V_r$ from sine wave fit radial velocities $V_r^{FSWF}$. In order to determine the latter the so called
filtered sine wave fit (FSWF) as a wind vector estimation technique introduced by Smalikho (2003) has been used . This
technique requires knowledge about the variance of "good" estimates which has been estimated based on the filter results
of MAD_part_I. An example illustrating intermediate results of the two-stage $MAD$ application is shown in Fig. 17. From
our experience we know that employing MAD_part_I particularly by means of the double MAD filter technique (Rosenmai,
2013) may contribute to save the azimuthal variability in $V_r$. Latter is important especially when the wind field was inhomoge-
neous and non-stationary during the 30 min measurement interval. This can be seen in Fig. 17 [b] illustrating the outcome of
MAD_part_I when applied to the intermediate results of **coarse filter II** for the 30 min type A measurement example shown



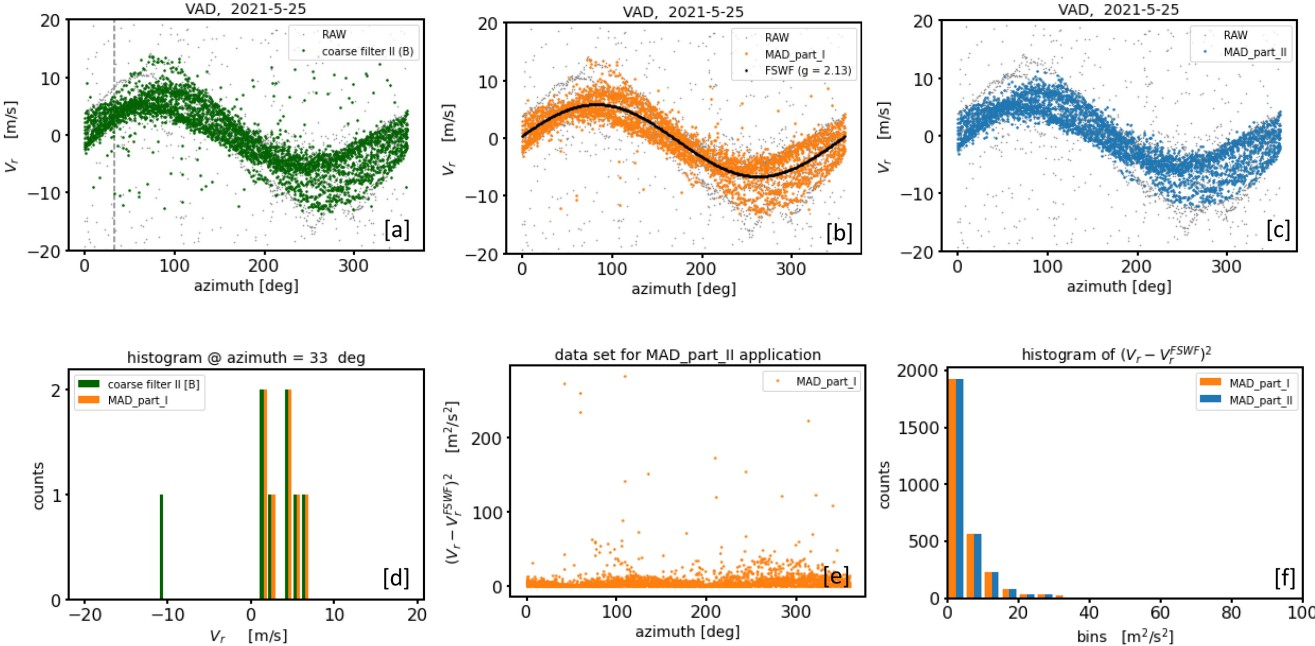

**Figure (17).** Intermediate results of the two-stage MAD filter applied to the outcome of **coarse filter II** for the type A measurement example shown in Fig. 16 [b]. Latter is shown here again in subfigure [a]. The outcome of MAD_part_I and MAD_part_II is shown in the subfigures [b] and [c], respectively. Furthermore in subfigure [b] the results of a filter sine wave fit (FSWF) are shown which has been calculated based on a standard deviation (here $\sigma = g = 2.13$ m s$^{-1}$) obtained from the colored data reflecting the results of MAD_part_II. Subfigures [d] - [f] show further details of each filter step such as an example for the azimuthwise application of the MAD filter technique at azimuth = 33° (see also the dashed vertical grey line in subfigure [a]), the data set representing squared deviations $(V_r - V_r^{FSWF})^2$ of radial velocity measurements $V_r$ from sine wave fit radial velocities $V_r^{FSWF}$ and the corresponding results of an application of the MAD filter technique to this data set.

in Fig. 17 [a] (see also Fig. 16 [b]). Here, relevant measurements reflecting the non-stationarity of the wind field still remain in the filtered data set even if they deviate substantially from the rest of the azimuthal dataset as can be seen for instance around the azimuth angles $\theta = 80°$ or $\theta = 250°$. Unfortunately, it is also noticeable in Fig. 17 [b] that after applying MAD_part_I not all "bad" estimates could be removed from the data set. This can be explained with the fact that often not enough data per azimuth

sector were available for a reliable calculation of the $median$ and $MAD$. For that reason MAD_part_II becomes necessary to further improve the "bad" estimate detection rate. Corresponding filter results are illustrated in Fig. 17 [c] and clearly show that the fraction of remaining "bad" estimates could be substantially reduced. Unfortunately, MAD_part_II also contributes to a severe cut-off of a huge fraction of directional variability which was actually possible to avoid by applying MAD_part_I. This can be attributed to the choice of the cutoff (here: $q = 3.5$) and shows very clearly the fundamental problems when using

statistical filter methods where cutoff values have to be carefully chosen and cannot be generalized as it would be required for a routine application.





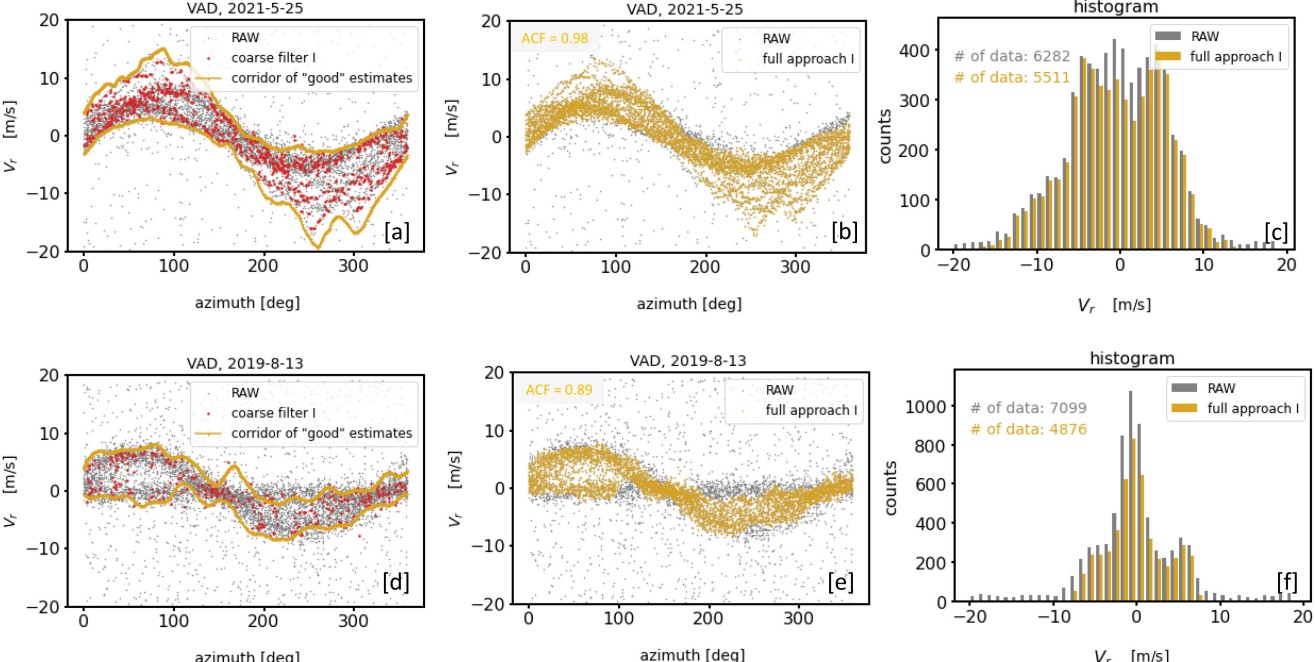

**Figure (18).** Results of the **filter for post processing** applied to the outcome of **coarse filter I** (see Fig. 14) for two measurement examples characterized by a different type of noise distribution (top: type A noise, bottom: type B noise). The subfigures in each row show from left to right: the identified borders of the area which define the corridor of "good" estimates ([a],[d]), the outcome of the re-activation step of previously discarded "good" estimates ([b],[e]) and the associated histograms which provide an overview of the final data availability of $V_r$ estimates identified as "good" data ([c],[f]).

### 4.3.2 Determination of the sinusoidal corridor of "good" estimates

The main advantage of **coarse filter I** in comparison to **coarse filter II** is the better performance w.r.t. the detection of "bad" estimates which makes the two-stage $MAD$ filter as a follow-up filter step of **coarse filter I** redundant at this point.

The disadvantage of **coarse filter I**, however, lies in the strong rejection of many obviously "good" estimates (see Sect. 4.2.1). We now describe a possibility to reverse individual data rejection decisions and therewith to increase the availability of reliable measurements again.

Recall that **coarse filter I** (see Fig. 14 [a] - [c]) provided a data set representing a suitable first guess for "good" estimates if visualized using the VAD perspective (see Fig. 14 [b]). Hence, the roughly filtered data can be used to narrow down the

sinusoidal area in the VAD space where most of "good" estimates can be found. This in turn offers the possibility to re-activate those radial velocities within the area boundaries as "good" data that have been discarded after applying **coarse filter I**. The outcome of such a re-activation as post processing step of **coarse filter I** is shown in Fig. 18. The identified borders of the area which define the corridor of "good" estimates shown in Fig. 18 [a] and [d] have been determined in the following two consecutive steps: Firstly, by calculating the previously accepted MIN/MAX radial velocity values for each azimuthal





direction. Secondly, by calculating the upper envelope of the MAX values and the lower envelope of the MIN values over the interval 0° to 360°. Then the re-activation of falsely rejected "good" estimates is done by considering all measurement points within the corridor defined by the upper and lower envelopes as "good" estimates. The corresponding results of this step are shown in Fig. 18 [b],[e]. Note that the procedure described above to determine the area of "good" estimates is relatively simple and has its weaknesses especially in case of low data availability which complicates the determination of the envelope due to a

small number of available MIN/MAX values. For such conditions a more sophisticated approach is needed. The re-activation results for the measurement example characterized by type A noise shown in Fig. 18 [b] match well with the data one would identify as "good" data from a visual point of view. The re-activation step is accompanied by a strong increase of reliable data compared to the outcome of **coarse filter I** (compare Fig. 14 [c] and 18 [c]). Hence the higher data availability achieved in this way may contribute to an improvement of the variance statistics required for a turbulence product retrieval. In contrast,

the re-activation step fails if applied to type B noise contaminated measurements which is shown in Fig. 18 [e]. This is due to both the poor first guess results for "good" estimates after applying **coarse filter I** which does not contain enough details to correctly narrow down the sinusoidal area of "good" estimates and some of the remaining "bad" estimates that belong to the specific class of "noise around zero". Note that to discard the latter, here we omitted the two-stage $MAD$ as a follow-up filter of **coarse filter I**. This is because we know from our experience that the generally significantly lower data availability of

reliable measurement data after the application of **coarse filter I** compared to **coarse filter II** turns out to be unfavorable for a successful $MAD$ application.

    So far the above introduced re-activation step of falsely rejected "good" estimates has been discussed in the context of a post-processing of filter results after applying **coarse filter I**. Even if an unjustified data loss of "good" estimates after an application of **coarse filter II** is not that substantial the above described re-activation step can also be applied to the outcome

of **coarse filter II**. However, due to the reasons mentioned in Sect. 4.3.1 this requires a previously executed two-stage $MAD$ filter. The corresponding results are shown in Fig. 19 where the significantly better results for the type B noise example (Fig. 19 [d] - [f]) are obvious. The disadvantage, however, is that with the re-activation step also a substantial number of "bad" estimates in the region around the reflection point of the sinusoidal corridor of "good" estimates is also assigned to the set of "good" data. At this point the corridor of "good" estimates and the horizontal band reflecting a higher concentration of

"noise around zero" overlap and no clear distinction between "good" and "bad" estimates is possible. This can also be seen by comparing the histograms shown in Fig. 16 [f] and Fig. 19 [f].

    Finally, it should be mentioned that with the reactivation of initially discarded data in the identified corridor of "good" estimates there is always a risk of returning a certain number of "bad" estimates if the raw measurements were contaminated with noise. Since "bad" estimates can be distributed over the whole measurement space of $\pm19$ m s$^{-1}$ they potentially also

occur in the corridor of "good" estimates. However, as long as the interest is only in mean wind and turbulence statistics which is primarily obtained using the VAD perspective the effect of such a small fraction of "bad" estimates is expected to be negligible.





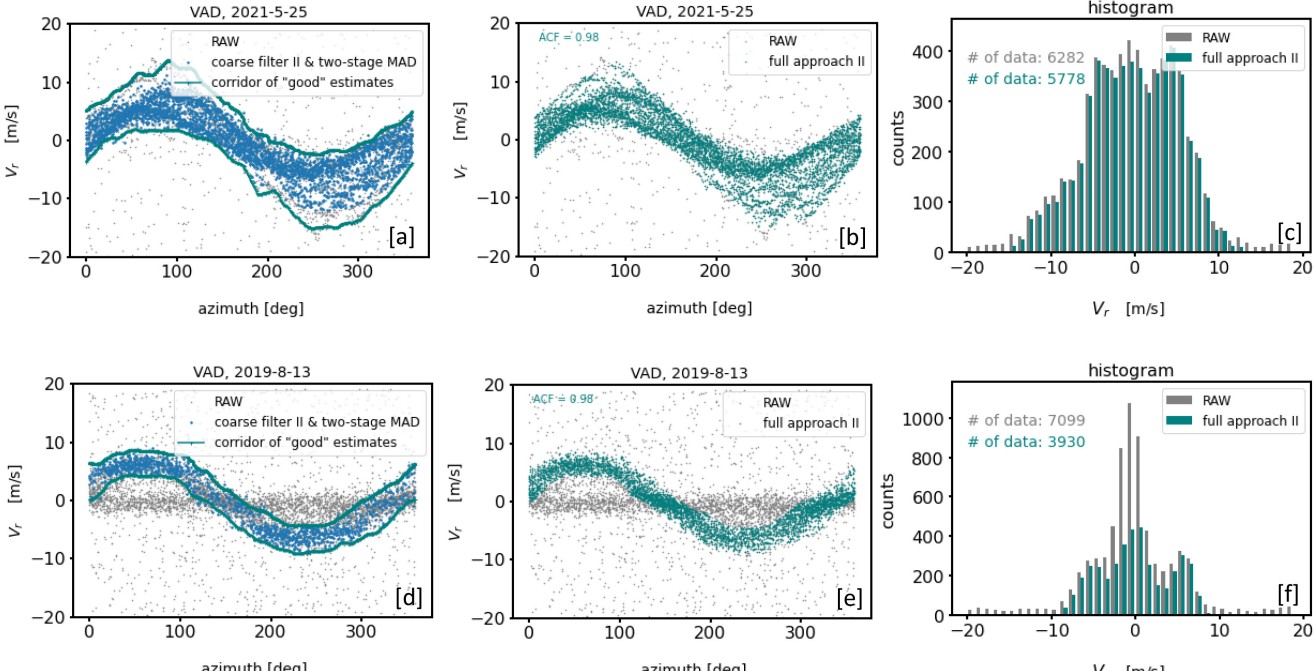

**Figure (19).** Results of the **filter for post processing** applied to the outcome of **coarse filter II** in combination with a follow-up two-stage MAD filter step (see Figs. 16 and 17) for two measurement examples characterized by a different type of noise distribution (top: type A noise, bottom: type B noise). The subfigures in each row show from left to right: the identified borders of the area which define the corridor of "good" estimates ([a],[d]), the outcome of the re-activation step of previously discarded "good" estimates ([b],[e]) and the associated histograms which provide an overview of the final data availability of "good" estimates ([c],[f]).

## 5   On the practical use of the new filtering techniques

In the previous subsections two different approaches, i.e. **approach I (= coarse filter I + filter for post processing)** and
**approach II (= coarse filter II + filter for post processing)**, to detect and reject "bad" estimates in DL radial velocity measurements from conical scans have been presented. Details of individual steps have been explained with focus on two examples which differed in terms of the observed noise distribution. Limits of usability depending on the type of noise have been discussed. The type of noise, however, is not the only factor affecting the applicability of the two different filtering techniques. Their success is also linked to the strength and temporal evolution of the wind during the measurement period.
The objective of this subsection is to provide a broader overview about filter results for a number of different measurement examples under different wind conditions. For reasoning of complexity, the focus is not on details of individual filter steps as in the foregoing sections but more on the final filter results of the full filter approaches. Knowledge and insights gained from this overview is used to develop a strategy to implement the filtering techniques for operational use.





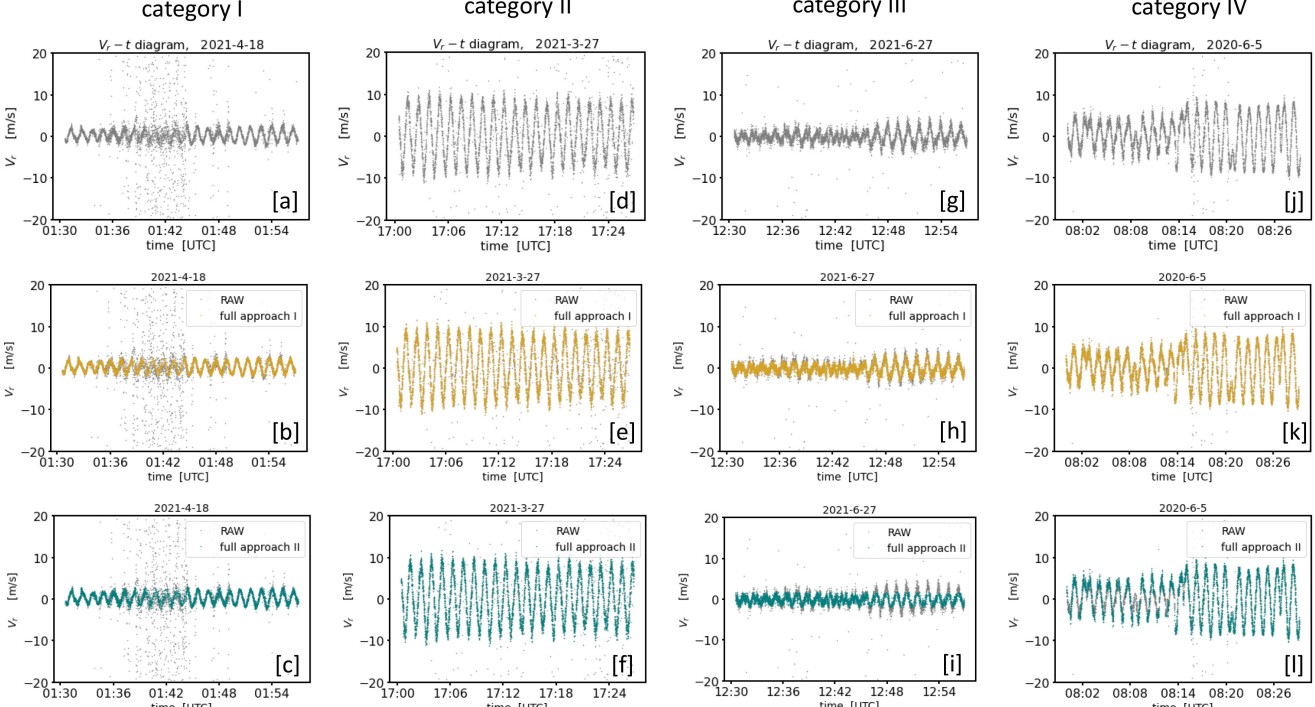

**Figure (20).** Overview about the final results of the two filtering techniques (**approach I** and **approach II**) for DL measurement examples contaminated with type A noise. The examples of each column represent four selected atmospheric conditions w.r.t. to the wind situation: *weak & stationary wind (category I)*, *strong & stationary wind (category II)*, *weak & non-stationary wind (category III)* and *strong & non-stationary wind* (category IV). The subfigures in the first row show the time series of the respective RAW data of the DL radial velocity measurements over a measurement interval of 30 min. In analogy the subfigures in the second (third) row show both the RAW data and the filter results of **approach I** (**approach II**) in different colors.

## 5.1 Intercomparison of approach I and approach II

For a systematic performance analysis of both filter techniques in different wind situations, four categories are distinguished: *(I) weak & stationary wind*, (II) *strong & stationary wind* , (III) *weak & non-stationary wind* and (IV) *strong & non-stationary wind.* A distinction between weak and strong winds is mainly motivated to show how differently the filter approaches perform if due to weak wind conditions "good" and particularly "bad" estimates representing "noise around zero" (type B noise) increasingly merge and a visual distinction is no longer possible (see also the *fourth* application note in Appendix D). A dis-

tinction between stationary and non-stationary measurement intervals is primarily motivated to show how differently the filter techniques perform if due to non-stationary effects the radial velocity distributions for certain azimuthal directions change from uni-modality to multi-modality. For both filtering techniques, i.e. **approach I** and **approach II**, the results from measurement examples of category I - IV are summarized in Figs. 20 - 21 for type A noise and in Fig. 22 for type B noise. From a first visual



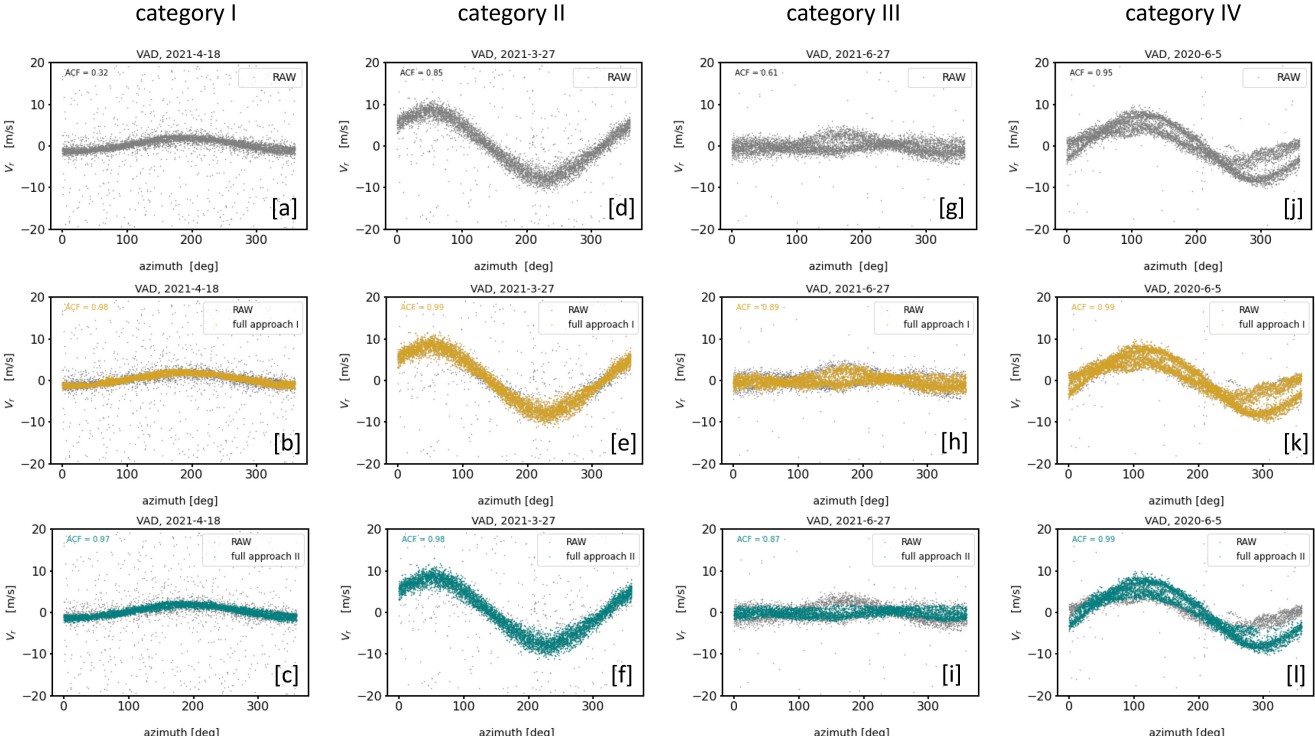

**Figure (21).** Same as in Fig. 20 except for the fact that the data are displayed using the VAD perspective.

comparison of **approach I** and **approach II** applied to measurements belonging to category I and category II it can be clearly

seen that there are no significant differences in the filter results provided the noise distribution is of type A.

An additional quantitative comparison of ACF values of the fully filtered time series of radial velocity measurements (see also Sect. 2.2.2) confirms these observations (see Fig. 21 [b], [e] for **approach I** and [c], [f] for **approach II**). Here the ACF values of the filtered time series range between 0.97 and 0.99 and indicate a low noise level in the filtered time series compared to the ACF values of the unfiltered data which range between 0.32 and 0.85 (Fig. 21 [a], [d]).

For measurements contaminated with noise of type B the performance of **approach II** (ACF ∼ 0.98-0.99) (see Fig. 22 [c], [f] and [i], [l]) is much better than the performance of **approach I** (ACF = 0.92) (see Fig. 22 [b], [e] and [h], [k]). This holds not only for strong but also for weak wind conditions. Concerning the latter this gives evidence that an overlay of "noise around zero" with the true signal does not in fact inhibit a successful application of filter **approach II** to separate "good" from "bad" estimates. However, it cannot be ruled out that some "noise around zero" is left as valid data close to the zero crossing of the sine

curve. The picture of an apparently better performance of **approach II** so far changes when comparing both approaches applied to measurements belonging to category III and category IV. Here, the performance of **approach I** (Fig. 21 [h],[k]) is better than the performance of **approach II** (Fig. 21 [i],[l]). This, however, is not because "bad" estimates have not been correctly detected by **approach II** but rather due to the wrong rejection of a substantial number of obviously "good" estimates. This





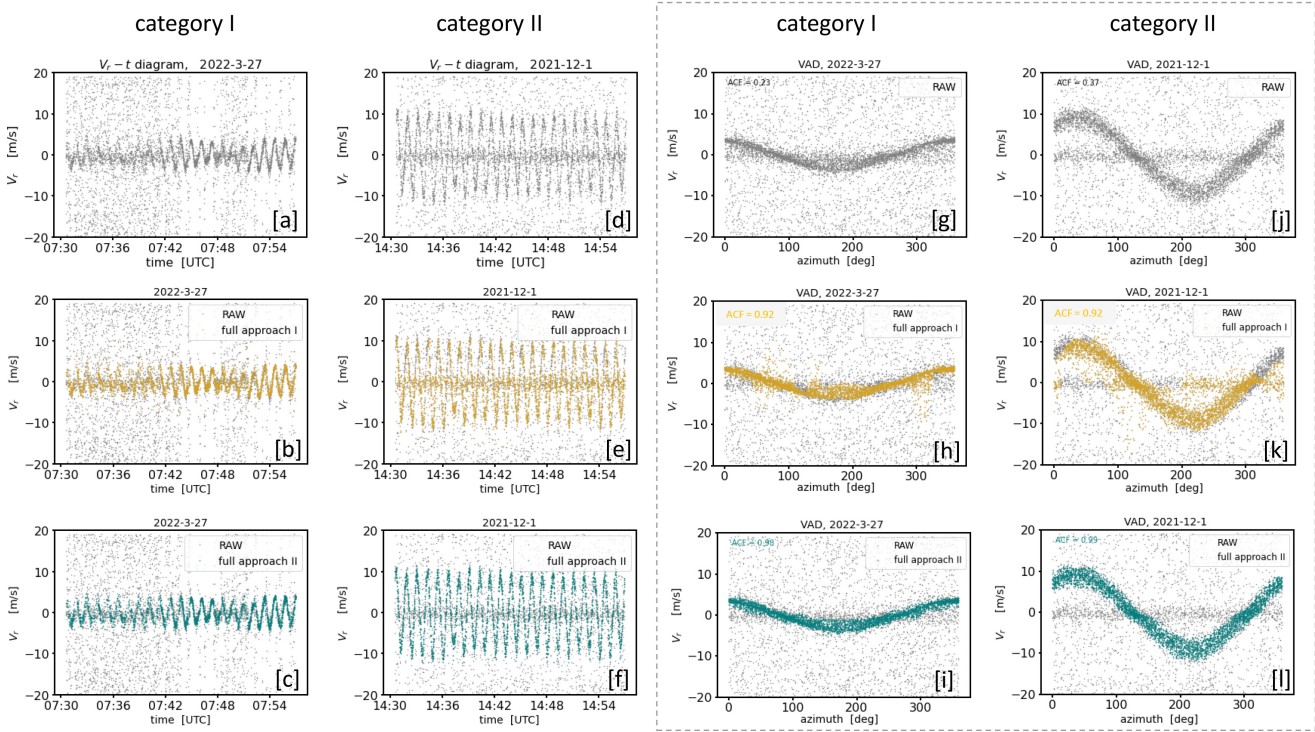

**Figure (22).** Same as in Fig. 20 for category I and II and in Fig. 21 for category I and II except that measurements contaminated with type B noise are considered. No situations of category III and IV could be found.

error can be traced back to a bimodal distribution in the measurement related $r_i$ - $count\{V_i\}$ diagram (not shown) which rather
reflects the non-stationarity of the measurement interval and which cannot be explained by the occurrence of "noise around zero" as discussed in Sec. 4.2.2 (see also Fig. 15). **Approach II**, however, is not designed to make this distinction and thus fails when applied to situations belonging to category III and IV. Finally it is worth to point out that weak and non-stationary wind conditions are for both approaches obviously difficult to manage, indicated by the comparably poor ACF values of the fully filtered time series in a range between 0.87 and 0.89. Note that examples for type B noise contaminated measurements of
category III and IV are not available.

A closer look at the filter results shown in Fig. 21 reveals one more thing calling for a comment. It becomes obvious that particularly during wind weak conditions (category I and III) the bands of "good" estimates sometimes seem too narrow from a purely visual point of view. This observation holds for both of the filter approaches (Fig. 21 [e], [g] and Fig. 21 [i], [k]). The reason for this lies in the method to determine the envelope in connection with the final re-activation step of previously
discarded "good" estimates (see Sect. 4.3.2). As a result, some information on the actual variability is lost in the filtered data set. This in turn may result in an underestimation of variances and hence the finally derived TKE. Evidence for the latter will be given in Sect. 5.3.





## 5.2 Strategy for implementation of approach I and approach II

The systematic intercomparison of both filtering techniques in the previous subsection leads to the following general conclu-
sion: For filter **approach I** the most limiting factor is "noise around zero" (type B noise) while problems may arise for filter
**approach II** due to non-stationarity in the wind field. These limiting factors are very different in nature. For **approach I** the
limitations are due to issues with the DL system's background noise. This seems to be an instrumental issue and there might be
a chance to get this problem solved by the manufacturer. In contrast, the limitations of **approach II** are that it cannot be used
for all wind situations. The DL end user has no influence on this and would therefore have to accept these limitations when
using the filter approach. Hence, from our point of view there are two options for an implementation of the filtering techniques
into an operational product retrieval process. Provided that the DL's background noise is always of type A, an implementation
of **approach I** appears to be sufficient to detect and reject "bad" estimates. If the DL system's background noise varies between
type A and type B noise both filtering techniques should be implemented. This should be combined with a decision strategy that
ensures the employment of **approach I** in case of measurements contaminated with type A (or no) noise or the employment
of **approach II** in case of type B noise. In order to be able to choose between the two options a good understanding of the
measurement systems' noise characteristics is required.

The combined application of both approaches presents another challenge for the implementation process. From a visual
perspective onto the measurement data it is easy to differentiate between type A and type B noise. However, for a routine
processing an automated decision-making strategy would be required. This could be arranged as follows: First one could
apply both filter approaches for each measurement interval under consideration. In doing so one obtains for each measurement
interval two differently filtered data sets which have not necessarily to be same. If one knows the "good" radial velocities, one
also knows the "bad" ones at the same time. Hence, with correct filtering the distributions of the "bad" estimates can provide
useful information about the type of noise occuring over the measurement interval. In case of type A noise one would expect
uniformly distributed data, whereas in case of type B noise a maximum close to zero would be characteristic (see also Sect.
2.2.1). Eventually, on this basis a decision on the appropriate filter method would be possible.

With the DL78 (see Sect. 2.1) that we used for our test measurements we had to deal with both types of noise. Hence, to
filter out "bad" estimates we had to implement both filtering techniques, i.e. **approach I** and **approach II**, taking additionally
the above discussed decision strategy into account. Against this background and for the sake of simplicity in the following we
will refer to this combined application of the different filtering techniques as **C**ombined **F**ilter using the notation **CF**.

## 5.3 Effect of the newly introduced filter on turbulence products

TKE retrievals obtained from differently pre-filtered DL input data using classical filtering techniques have been shown in Fig.
9 while comparing them with independent sonic data at 90 m height. TKE retrievals based on **CF** pre-filtered input data have
been analysed in a similar way. The results are shown in Fig. 23. A comparison of the diagrams in Fig. 23 [a] - [d] with the
diagrams in Fig. 9 [i] - [l] showing a TKE retrieval based on CNS pre-filtered data reveals first few signs of an improvement
regarding one important aspect: The significant underestimation of larger values in TKE retrievals based on CNS pre-filtered





**Figure (23).** Comparison of TKE from DL at 95 m height with TKE from sonic at 90 m height over the period from 5-18-2021 to 7-17-2021. The DL TKE has been calculated as proposed by Smalikho and Banakh (2017) where the input data have been pre-filtered using the CF filtering technique. The subfigures represent comparisons between DL and sonic data using different visualization techniques, i.e. scatterplots [a], a Bland Altmann plot [b], time series plots (one day (5-25-2021), only) [c], and histograms [d]. The abbreviations "QC: lev_x" refer to different levels of product quality.

input data no longer occurs in TKE retrievals based on **CF** pre-filtered input data. This contributes to a slightly improved statistical agreement with sonic data, i.e., $R^2 = 0.98$ and $RMSD = 0.23$ m$^2$ s$^{-2}$ for TKE values based on **CF** pre-filtered input data (Fig. 23 [a]) compared to $R^2 = 0.97$ and $RMSD = 0.27$ m$^2$ s$^{-2}$ if the input data set is pre-filtered using the CNS (Fig. 9 [i]). The improvement of TKE retrievals becomes also evident by a better detection of atmospheric situations with pronounced weak and strong TKE in the course of the day (Fig. 9 [j], Fig. 23 [c]) and the fact that for larger averaged TKE






**Figure (24).** Same as in Fig. 10 except that instead of the CNS pre-filtering the CF pre-filtering has been used.

values (e.g. $0.5\,(\text{TKE}_{DL} + \text{TKE}_{Sonic}) > 2\ \text{m}^2\ \text{s}^{-2}$) the distribution of the data points in the Bland and Altmann diagram gives no indication for an underestimation of TKE retrievals based on a **CF** pre-filtering (Fig. 23 [b]) compared to a CNS pre-filtering (Fig. 9 [k]). Concerning the aspect of data availability a comparison of the histograms in Fig. 23 [d] and in Fig. 9 [l] reveals a slight increase of the number of reliable data for TKE retrievals based on **CF** pre-filtered input data (83%) against those based on a CNS pre-filtering (79%).

The first signs of an improved performance of the **CF** filter over the CNS filter can be further substantiated when comparing TKE values from larger measurement heights. As already discussed in Sect. 3.3 the robustness of a filtering method is revealed particularly at larger heights where the SNR values are lower and therewith the probability of "bad" estimates is higher. Hence,





in analogy to Fig. 10, a comparison of TKE values based on **CF** pre-filtered input data with those based on the SNR thresholding

technique using $\mathrm{SNR}_{thresh}$ = -12.7 dB, is shown in Fig. 24. Recall that latter were used as an alternative reference for sonic data at larger measurement heights. When comparing the diagrams the advantages of the new filter approach become more obvious. There is no underestimation of larger TKE values when the **CF** filtering technique has been used. This can be seen in the scatterplots shown in Fig. 24 [a], [c] but it is most pronounced in the Bland Altmann diagrams shown in Fig. 24 [b], [d]. Furthermore, there are much less data with a strong overestimation in the range of intermediate TKE values based on **CF**

pre-filtered input data. These results lead to a significant improvement in the limits of agreement. If the upper (lower) limits were 37.63 % (-38.52 %) for TKE retrievals based on CNS pre-filtered input data (Fig. 10 [d]), they reduce to 10.25 % (-17.17 % ) for TKE retrievals based on **CF** pre-filtering (Fig. 24 [d]). Note that these numbers also imply a better agreement between TKE values based on $SNR_{thresh}$ pre-filtered input data with those based on **CF**. Due to the differences in the measurement methods, i.e. conical scan vs. point measurement, this aspect is not that obvious when comparing the respective DL TKE values

with sonic data. Additionally it is important to note that the Bland and Altmann diagrams in Fig. 24 confirm the systematic underestimation (bias) of TKE that is also noticeable when comparing the TKE values based on **CF** pre-filtered input data with sonic data as shown in Fig. 23 [b]. A possible explanation for this underestimation is given at the end of Sect. 5.1. We assume that with a more sophisticated method for the determination of the envelope of "good" estimates in connection with the final re-activation step of previously discarded "good" estimates (see Sect. 4.3.2) such a bias could be avoided. Finally, it is worth

to point out that according to the decision strategy discussed in Sect. 5.2 the decision was on **approach I** in about 90% of the considered time intervals which implies that the "noise around zero" issue was relatively rare over the measurement period 5-18-2021 to 7-17-2021.

## 6   Summary

First test measurements for a desired routine application of DL turbulence measurements based on the approach outlined

in Smalikho and Banakh (2017) revealed unforeseen difficulties concerning the quality of radial velocity estimates. These difficulties turned out to be the consequence of the specific requirements on the scanning strategy and the associated limitation of pulse accumulations feasable for the measurements. During a 24/7 application over a several-months period and therewith a naturally varying aerosol load in the atmosphere, this limitation frequently became obvious through comparably weak values for the SNR and an increased fraction of "bad" estimates (random outlier; noise) in the measurements. If not properly filtered,

"bad" estimates can contribute to large errors in the retrieved turbulence variables. This raised the issue of an appropriate noise filtering, i.e. a method that can be used to separate "good" from "bad" estimates in a series of radial velocity measurements, prior to a turbulence product retrieval. Looking for a suitable noise filtering method, first of all differences in traditionally used filtering techniques have been worked out and respective advantages and disadvantages were discussed. Using the example of the well established SNR - thresholding technique a literature-based overview of the different possibilities of an SNR - threshold

determination was given. In this context a selection of theoretical approaches taking into account the number of accumulated pulses for a radial velocity estimate was verified, based on various measurement examples with different pulse accumulations.





In the practical application the approximate equation by Abdelazim et al. (2016) turned out to be the most appropriate one if a complete removal of all "bad" estimates is essential. However, during the verification it also became clear that the strong increase of the SNR - threshold value with decreasing pulse accumulations significantly reduces the availability of reliable ("good") radial velocity estimates. This would strongly limit the derivation of turbulence variables and thus the intended routine application. In contrast to the SNR - thresholding technique the advantage of the CNS approach was confirmed, namely the higher data availability after the filtering process. The quality of the filtered time series, however, was often not satisfactory for a turbulence retrieval. Two causes could be identified for this: (1) an a priori assumption about the radial velocity variance as a prerequisite for the application of the CNS method and (2) an application limitation to DL measurements characterized by uniformly distributed noise (i.e. white noise). The first point is critical, since turbulence measurements essentially rely on variance measurements. It has been shown that inappropriate assumptions either reject too many "good" estimates or leave to many "bad" estimates. Particularly this is the case when the a priori assumed variance is either too small or too large compared to the true atmospheric situation. As a result, errors in the retrieved turbulence variables occur, as for instance an under- or overestimation in the TKE which has been shown by comparing the TKE with independent reference measurements. The second point mentioned above is a serious limitation as the noise distribution in DL measurements with low pulse accumulation not always represents white noise. This holds at least for StreamLine DL systems from the manufacturer HALO Photonics which have been used for the test measurements in this study. During our test measurements a second type of noise was identified showing a pronounced aggregation of noise values around zero. To our knowledge, this type of noise has not yet been described and analyzed in the literature. Overall, we finally came to the conclusion that the filtering techniques available so far were not appropriate to be used in a pre-processing step to generate noise-free data that can serve as a suitable input for the derivation of turbulence variables.

The drawbacks of the frequently used filtering techniques motivated our work to seek new ideas for filter methods that can be applied to noise contaminated measurements from conically scanning DL systems with low pulse accumulation. They should allow for both an accurate noise filtering and largest possible data availability. Two different approaches (I + II) were pursued in order to account for possible emerging differences in the noise distribution. Their basic structure consists of two parts, namely a coarse filter and a so-called filter for post processing. Although each approach has a different coarse filter (I + II) they are applied in both cases based on a newly introduced framework of the VV90D perspective. By plotting the time series of radial velocity measurements (V) from conically scanning DL against the same measurement series but with a phase shift by 90 degrees (V90) the graph of noise free DL measurements shows distinctive circular patterns which are increasingly faint, the more noisier the data are. Using this perspective on the measurement data, coarse filter I works by the identification of "bad" estimates in terms of singular points occurring in subsets of the full measurement series which are confined to different circular rings of radius R and fixed width in the VV90 plane. Coarse filter II exploits the fact that the autocorrelation function can provide valuable information about the general existence of "bad" estimates in noise contaminated measurements. The filter works by means of an iterative consideration of circular rings with increasing width in the VV90 plane and the calculation of the autocorrelation function of the measurement data that can be found in these rings. Within the iterative process circular rings with mostly good data can be easily located, if the autocorrelation function is used to define a termination criterion. Generally,



both coarse filter give a useful first guess about "good" estimates. Depending on the type of noise, either coarse filter I or coarse filter II shows a better performance w.r.t. data availability and reliability of the data which were assessed as "good" in this way. The filter for post processing is applied by considering the DL measurements using the well known VAD perspective, where

reliable measurements from a number of subsequent conical scans typically describe a sinusoidal band. Based on the first guess information about "good" estimates from the coarse filter (I + II) the filter for post processing was developed to determine the envelopes of this sinusoidal band and thus to further narrow down the whole area of "good" estimates.

The results obtained with both newly introduced filter approaches were qualitatively and quantitatively verified. While the qualitative verification was based on a purely visual assessment of the filter results, the quantitative verification was based

on an evaluation of the TKE that was calculated using the filtered measurements as input data for the turbulence retrieval. Because still included "bad" estimates after the filtering process would introduce large errors in the final TKE product this is an indirect way to verify the filter results. It could be shown that the deficiencies in the filtered time series and the related problems regarding data availability and quality of derived turbulence variables emerging by an application of traditional filter methods have been significantly reduced with the new filter approach. In this way, we have found a solution to deal with noise

contaminated DL measurements if low pulse accumulations for the radial velocity estimates are used. Therewith we have also created a basis to be able to use the turbulence retrieval as outlined in Smalikho et al. (2017) for a 24/7 routine application.

This new filter method can also be used beyond the application described here generally to other DL applications using conical scans. One example could be DL wind gust retrievals based on a scan mode as described in Steinheuer et al. (2022). Their mode also uses a small number of pulses (3000 pulses/ray) to provide high temporal resolution, which is necessary for

the derivation of wind gusts which are defined based on 3 s running mean wind data (World Meteorological Organization, 2018). It remains for future work to apply this new filter for the wind gust retrieval.

*Data availability.* Doppler lidar data sets used for the analysis including radial velocity measurements (level 1 data) and retrieved wind and turbulence products (level 2 data) are available via the ZFDM Repository of the Universität Hamburg (https://doi.org/10.25592/uhhfdm. 10559).

**Appendix A: StreamLine DL custom scan files**

For the configuration of the Doppler Lidar to measure turbulence variables as proposed by Smalikho and Banakh (2017), two files (*.txt, *.dss) are required. These were created under the guidance of the User Manual for Streamline Scanning Doppler LiDAR System (Revision 04 | # DOC 0004-01355 by Lumibird) and have the following content:

*routine.txt*
$A.1 = 50, S.1 = 694, P.1 = 0 * A.2 = 50, S.2 = 5000, P.2 = -24514$
$W0$
$A.1 = 50, S.1 = 694, P.1 = 12000000 * A.2 = 50, S.2 = 5000, P.2 = -24514$





$W\,80000$

$A.1 = 50, S.1 = 694, P.1 = 0 * A.2 = 50, S.2 = 5000, P.2 = -24514$

$W\,0$

*routine.dss*

| 000000 | routine | 2 | C | 0 |
|--------|---------|---|---|---|
| 010000 | routine | 2 | C | 0 |
| 020000 | routine | 2 | C | 0 |
| ⋮      |         |   |   |   |
| 220000 | routine | 2 | C | 0 |
| 230000 | routine | 2 | C | 0 |

First a configuration file *routine.txt* is created, which defines all information about acceleration (A), speed (S) and position (P) of the two motors of the DL scanner and, if necessary, wait times (W). More detailed explanations can be found in section

6.4.2 of the user manual. For the operation of the Doppler lidar, the scan scheduler is set to use a daily scan file *routine.dss*, in which the *routine.txt* file is used (column 2). Additionally, start times of measurements (hhmmss), k/samples per ray, whether the scan is of step/stare (S) or CSM (C) type, and the focus position must be specified. A detailed explanation of the use of the daily scan schedule can be found in section 6.4.3 of the user manual.

**Appendix B: Radial velocity uncertainty estimates**

An equation for uncertainty estimates of DL radial velocity is given in Pearson et al. (2009) and reads:

$$\sigma\,(m\,s^{-1}) \;=\; 2\,(\pi^{0.5}/\alpha)^{0.5}\,(1+0.16\alpha)\,(\Delta\nu/N_p^{0.5}) \tag{B1}$$

with

$$\alpha \;=\; SNR/[(2\pi)^{0.5}(\Delta\nu/B)] \tag{B2}$$

$$N_p \;=\; M\,N_a\,(SNR) \quad . \tag{B3}$$

Here, the system parameters $N_p, B, M$ denote the accumulated photocount, the used bandwidth and the gate length in points, respectively. The signal spectral width $\Delta\nu$ depends on both instrumental and atmospheric conditions Doviak and Zrnic (1993), i.e.,

$$\Delta\nu \;=\; \sigma_{tot} \;=\; \sqrt{(\sigma_i)^2 + (\sigma_a)^2} \quad , \tag{B4}$$


with $\sigma_i$ given through (please see eqns. (6) and (7) in Frehlich (2004)

$$\sigma_i = \lambda\,\omega/2 \tag{B5}$$

$$\omega = \frac{(\ln 2/2)^{1/2}}{\pi\Delta t} = \frac{0.1873906}{\Delta t} \quad , \tag{B6}$$

where $\lambda$ denotes the wavelength of the Doppler Lidar. Here, $\Delta t$ denotes the pulse width which is used to calculate the spectral width $\omega$ and which can be transformed into a spectral width $\sigma_i$ in velocity space via eqn. (B5). Note that in Pearson et al. (2009) not the pulse width $\Delta t$ but the pulse length $\Delta r$ is given as a Lidar parameter. By knowledge of $\Delta r$ the pulse width $\Delta t$
can be calculated via

$$\Delta r = c\,\Delta t/2 \quad , \tag{B7}$$

where $c$ denotes the speed of light (Frehlich, 2004)). The value $\sigma_a$ in B4 denotes the atmospheric broadening factor. In Pearson et al. (2009) it is assumed that $\sigma_a = 1\ m\ s^{-1}$.

**Appendix C: On the determination of the critical radius $r_c$**

Using the framework of the VV90D perspective, the critical radius $r_c$ can be determined by analyzing DL measurements from higher range gate numbers where only background noise and no true signal characterizes the DL measurements. An example for such a situation is shown in Fig. C1. Here the circular rings with central radii $r_i$ below $r_c = 4.5$ show an increased availability of data which can be clearly assigned to "noise around zero". For $r_i > r_c$ the data availability is almost uniformly distributed. We know from our experience of working with various DL systems that the noise characteristics can be different.
For that reason the above described features are characteristic for the DL used in our studies and cannot be generalized.

**Appendix D: Guidance on a practical implementation of coarse filter II**

From a practical point of view there are four further issues which are worth to be pointed out for DL users when applying **coarse filter II**: *Firstly*, from our experience we know that for an azimuthal resolution of $\Delta\theta = 1°$ a value of ACF $(\tau = 1) <$ 0.95 indicates a relatively high fraction of "bad" estimates (see also the examples shown in Fig. 1). Hence for a successful
application of the method the threshold value ACF$_{thresh}$ = 0.95 is recommended. For DL measurements with lower azimuthal resolution than $\Delta\theta = 1°$ this value can be different (see Sect. 2.2.2). *Secondly*, it can be useful to apply **coarse filter II** a second time for a further improvement of the filter results. Prior to a second application, however, the diagrams representing the VV90D perspective have to be re-drawn based on the filtered time series obtained from the first application of **coarse filter II** but with a phase shift in the opposite direction, i.e. $\Delta\Theta = +90°$. The outcome of a second application of **coarse filter II**
(indicated by capital letter B in the plot legends) in comparison to the results obtained from the first application (indicated by capital letter A in the plot legends) is shown in Fig. 15 and Fig. 16. In these cases, however, the second application (i.e. coarse filter II (B)) has no significant advantage compared to the first application (i.e. coarse filter II (A)) since the ACF values for



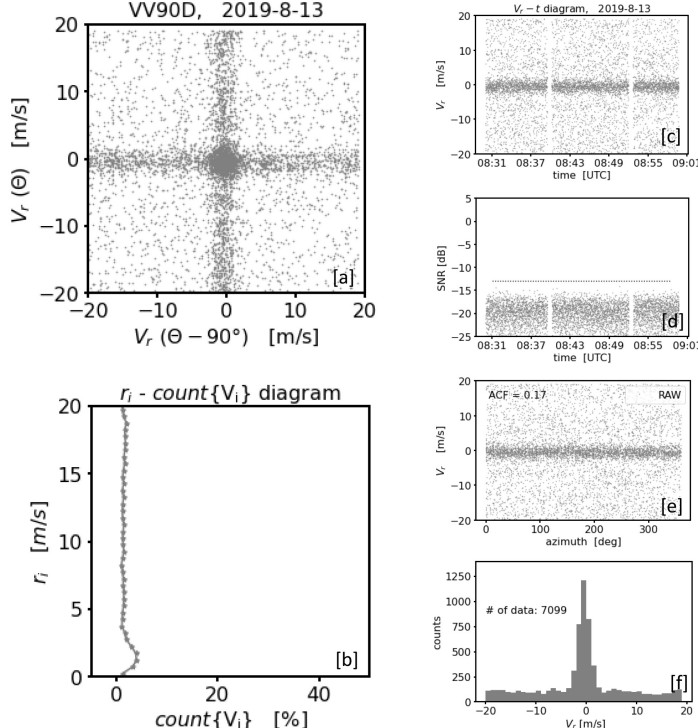

**Figure (C1).** Noise characteristics of the Doppler lidar DL78. The data represent a 30 min time interval at range gate number 90. The subfigures on the left show: [a] the VV90D perspective of the radial velocity measurements and [b] the corresponding $r_i$ - $count\{V_i\}$ diagram to describe the availability of data in different circular rings $r_i - \Delta r \leq r_i \leq r_i + \Delta r$ spanning the (V90,V) plane where $i = 1, .., n$ with $n \in \mathbb{N}$ and $\Delta r = 0.5$ m s$^{-1}$. The subfigures on the right are with regard to their order analogous to the plots shown in Fig. 1, i.e. [c] the time series plot of the DL radial velocity estimates, [d] the corresponding time series plot of the signal-to-noise ratio (SNR), [e] the corresponding VAD plot of the measurements and [f] the histogram of DL radial velocity estimates.

each iteration step (see Fig. 15 [c],[d]) remain above the ACF threshold value. Hence, there is no need to discard subsets $V_i$ from the circular rings $r_i - \Delta r \leq r_i \leq r_i + \Delta r$ under consideration. Note, however, that although no data have been discarded

the final filter results after the second application (see Fig. 16) show a reduced data availability in comparison to the first application. This effect rather is a technical consequence of the opposite shift of the filtered time series obtained based on the first application. *Thirdly*, there are also practical limits in the application of **coarse filter II**. They arise when for the initial circular ring with a maximum number of data the value of the data count is already very low. This is especially the case when the original time series is highly contaminated with noise. In this case the gaps due to flag values of the initially

determined time series are too large so that no meaningful results are obtained with a linear interpolation to fill these gaps. Accordingly, the corresponding ACF value of the time series is not suitable to be used as a trustworthy indicator for the occurrence of "bad" estimates. Since it is difficult to give a threshold value for the required frequency maximum to exceed in order to get reliable ACF results, it is recommended to always check the ACF of the finally filtered time series and to use





only those filtered series where the condition ACF > ACF$_{thresh}$ is met. *Fourth*, in the case of a pronounced type B noise

it may happen, that the data availability in those circular rings including most of "noise around zero" is comparable to the availability of data in those rings including most of the true signals, i.e. "good" estimates (see Fig. 13 [e] ). The challenge in such a case is to ensure that for the generation of the initial filtered time series the circular ring $r_i - \Delta r \leq r_i \leq r_i + \Delta r$ with $MAX(count\{\mathsf{V}_i\})$ has been chosen correctly, i.e. the subset of high data availability with the "good" signals instead of the "bad" signals. To achieve this it may be helpful to always test the data availability distribution per circular ring (i.e. the

distribution displayed in the $r_i$ - $count\{\mathsf{V}_i\}$ diagram) on multi-modality. When multi-modality occurs and a circular ring with a central radius $r_i < r_c$ representing a local maximum of data does exist, all circular rings with central radii $r_i < r_c$ should be excluded prior to employing **coarse filter II**. Here, $r_c$ denotes a critical radius below which one has to expect a pronounced concentration of "bad" estimates if the noise is of type B. How this value can be determined is described in more detail in Appendix C. For the DL used in our studies we found $r_c = 4.5$. This value is also relatively constant with time and explains the

missing data in the $r_i$ - $count\{\mathsf{V}_i\}$ diagram shown in Fig. 15 [e] for $r_i < 4.5$ compared to the $r_i$ - $count\{\mathsf{V}_i\}$ diagram shown in Fig. 13 [e] for the same measurement interval. It should be noted, however, that the above described procedure of circular ring exclusion is not recommended for type B contaminated measurements during weak and stationary wind conditions. In such a case multi-modality in the $r_i$ - $count\{\mathsf{V}_i\}$ diagram is not expected either, since with decreasing wind speeds circular rings including most of the true signals and circular rings including most of the noise around zero increasingly merge, it carries the

risk to discard a huge fraction of "good" estimates. The latter would negatively affect the possibility to derive reliable wind and turbulence values during weak wind conditions.

*Author contributions.* CD and EP performed the measurements. EP conceived the investigations and did the formal analysis of the data. EP developed the filter methods. EP implemented the filter methods in continuous interaction with CD. CD investigated the transferability of the filter methods to other scan strategies. EP visualized the data and wrote the manuscript draft. CD and EP discussed and finalized the paper

together.

*Competing interests.* The authors declare that they have no conflict of interest.

*Acknowledgements.* We thank Ronny Leinweber for his support in configuring the Doppler lidar and creating the level-1 data. We thank Markus Kayser for introducing the idea of a coordinate transformation of VAD data into the phase-space perspective. Frank Beyrich is acknowledged for valuable contributions to the final writing of the manuscript. Thanks to Volker Lehmann for comments on the manuscript.





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
