# Peer review of "Noise filtering options for conically scanning Doppler LiDAR measurements with low pulse accumulation"

_Atmospheric Measurement Techniques, 2023_

## Referee Comment (RC1)

**Review of the manuscript amt-2023-153 entitled "Noise filtering options for conically scanning Doppler LiDAR measurements with low pulse accumulation" by E. Paschke and C. Detring.**

This manuscript proposes revised filtering strategies of Doppler LiDAR data collected through the VAD technique with low pulse accumulation with the aim of retrieving wind turbulence statistics, such as TKE.

The manuscript begins with a roughly comprehensive Introduction, indicating clearly the work by Smalikho and Banakh (2017) as a reference for this work. In Sect. 2, the experimental setup is described followed by a characterization of typical noise encountered in LiDAR measurements. In Sect. 3, a review of some of the filtering techniques for LiDAR data is provided. Subsequently, Sect. 4 describes the filtering techniques proposed, which is followed by some applications of these filters in Sect. 5.

As a Doppler LiDAR researcher, I enjoyed reading this manuscript where the authors share their experience in filtering and processing VAD Doppler LiDAR data. I believe this manuscript will provide good guidelines in that realm, especially for younger researchers and practitioners. I find the filtering techniques proposed very reasonable and hopefully effective for producing noise-reduced LiDAR data. The main comment I have is about the length of the manuscript. I believe the manuscript can be significantly shortened by reducing lengthy, not strictly necessary discussions, and intermediate summaries and recaps provided throughout the manuscript. I think a shorter and more focused manuscript will enhance its impact. Please find below some details comments, which might help to revise the manuscript.

**Detail Comments:**

1.      L28 – Please add some references for works related to the various turbulence parameters in order to provide sufficient information on the retrieval procedures adopted.
2.      L54 – "… increased level of noise…" with respect to what condition? Reducing accumulation time? Please clarify.
3.      L149 – Please clarify how these noise-free measurements are obtained at this stage.
4.      L271 – "A more simpler", please revise this typo.
5.      Sect. 3.2 – For the sake of completeness, it would be good if the authors could summarize the procedures for the retrieval of TKE from the VAD lidar data for both works SM2017 and KR1986.
6.      L367-369 – You can briefly summarize the procedure used to retrieve TKE from the sonic anemometer.
7.      L481 – I am not sure you introduced the acronym VV90D. Please verify.
8.      Fig. 11 – I believe the reader does not have all the information needed to understand Fig. 11, e.g., the two-stage MAD filter. I would suggest removing this figure because the description provided in the text is already sufficient.
9.      L492 – at this stage, it is not clear why the authors propose a shift of 90 deg. Please homogenize it with the text.
10.     Fig. 12 – Letters [c] and [f] are missing in the caption
11.     L631 – Please revise "on the one hand".

12.    Eq. 9 – Can you please explain the origin of the coefficient 0.6745?
13.    L744-749 – You can remove this summary of the previous section.

---

## Author Comment (AC1)

**Answer to Referee #1**

**Review of the manuscript amt-2023-153 entitled "Noise filtering options for conically scanning Doppler LiDAR measurements with low pulse accumulation" by E. Paschke and C. Detring**

We would first like to thank the referee for reviewing our manuscript, the overall positive assessment of our work, and for the constructive comments. Below we will respond to the comments and point out changes we made as we revised our manuscript. The reviewer's comments are in black italic; our responses are in blue.

*This manuscript proposes revised filtering strategies of Doppler LiDAR data collected through the VAD technique with low pulse accumulation with the aim of retrieving wind turbulence statistics, such as TKE.*

*The manuscript begins with a roughly comprehensive Introduction, indicating clearly the work by Smalikho and Banakh (2017) as a reference for this work. In Sect. 2, the experimental setup is described followed by a characterization of typical noise encountered in LiDAR measurements. In Sect. 3, a review of some of the filtering techniques for LiDAR data is provided. Subsequently, Sect. 4 describes the filtering techniques proposed, which is followed by some applications of these filters in Sect. 5.*

*As a Doppler LiDAR researcher, I enjoyed reading this manuscript where the authors share their experience in filtering and processing VAD Doppler LiDAR data. I believe this manuscript will provide good guidelines in that realm, especially for younger researchers and practitioners. I find the filtering techniques proposed very reasonable and hopefully effective for producing noise- reduced LiDAR data. The main comment I have is about the length of the manuscript. I believe the manuscript can be significantly shortened by reducing lengthy, not strictly necessary discussions, and intermediate summaries and recaps provided throughout the manuscript. I think a shorter and more focused manuscript will enhance its impact. Please find below some details comments, which might help to revise the manuscript.*

We understand the criticism about the length of the manuscript. We tried to reduce the length of the manuscript by reorganizing its structure, shortening of individual text passages and reducing the number of illustrations. In particular, significant changes have been made to sections 3 and 5. As part of the restructuring, a stronger focus was sought on the filter methods, which now takes up the main part of the manuscript on pages 12 - 26. The advantages of the newly introduced filtering techniques compared

to common filter techniques are then discussed using the example of a special application (TKE retrieval) on pages 27 - 32. Overall, the number of pages in the main part of the manuscript was reduced from 46 pages to 34 pages. At the same time, the number of pages in the appendices increased from 4 to 10.

**Detail Comments:**

*1. L28 – Please add some references for works related to the various turbulence parameters in order to provide sufficient information on the retrieval procedures adopted.*

We have slightly revised the text here. In fact, so far only DL-based wind profile measurements are routinely carried out at MOL-RAO. DL-based turbulence measurements based on the retrieval approach introduced by Smalikho and Banakh (2017) are the planned next step. The work submitted here represents a part of the necessary preliminary studies. There is at present no further experience with other retrieval methods for DL-based turbulence measurements at MOL-RAO.

*2. L54 – "... increased level of noise..." with respect to what condition? Reducing accumulation time? Please clarify.*

Yes, the higher noise level can be attributed to the reduced accumulation time. In text lines 57-68 of the original manuscript, this point is discussed in more detail as a result of a scanning strategy with high temporal and spatial resolution. In the revised manuscript (line 58-59), the text has been adjusted to address this issue more directly.

*3. L149 – Please clarify how these noise-free measurements are obtained at this stage.*

All three measurement examples in Fig. 1 have been taken with the same DL system with identical configuration (e.g. Na = 2000 pulse accumulations). Despite the low pulse accumulations there are measurement cases with and without noise. This can be explained by the natural variability in the atmospheric aerosol content over the course of a day and with altitude. Aerosols act as backscattering targets and their atmospheric loading influences the quality of the DL signals and therewith the amount of noise in the measurements. A sufficiently large amount of aerosol can contribute to noise-free DL measurements even for low pulse accumulations. Little aerosol combined with low pulse accumulation, however, represent an unfavorable constellation for achieving good data quality. (Note that this explanation has been additionally introduced in the revised text; see lines

157 - 163.)

4. *L271 – "A more simpler", please revise this typo.*

Done.

5. *Sect. 3.2 – For the sake of completeness, it would be good if the authors could summarize the procedures for the retrieval of TKE from the VAD lidar data for both works SM2017 and KR1986.*

The length of the manuscript was criticized and calls for significant reductions were made. We think that an additional summary of both the Kropfli (1986) approach and the Smalikho and Banakh (2017) approach would contradict this request. Moreover, the focus of the present manuscript is clearly set on the noise filtering, and the TKE retrieval just serves as an example for an application of a scanning strategy that requires new ways of noise filtering. We thus feel that providing the references here would be sufficient in this context. Please also note that with the revision of the manuscript Section 3.2 (old manuscript version) was moved to the appendix (new version; Appendix H).

6. *L367-369 – You can briefly summarize the procedure used to retrieve TKE from the sonic anemometer.*

Done. Due to the restructuring of the manuscript, this information can now be found in line 560-563 of the revised manuscript.

7. *L481 – I am not sure you introduced the acronym VV90D. Please verify.*

The abbreviation VV90D has been introduced in analogy to the VAD. Latter is an acronym for **v**elocity-**a**zimuth-**d**isplay which describes a diagram with these axis labels. The notation VV90D was chosen in analogy to the VAD plot. However, VV90D does not represent an acronym formed from the first letters of several words. Although V represents the velocity, V90 represents the same velocity time series shifted by 90 degrees. Unfortunately, the latter cannot be summarized in one word. In order to give the reader a quick solution to what lies behind it, reference is immediately made to the following section with further details (line 280 of the revised manuscript). Note that in the revised manuscript the acronym VV90D is introduced in line 277 instead of line 481.

8. *Fig. 11 – I believe the reader does not have all the information needed to understand Fig. 11, e.g., the two-stage MAD filter. I would suggest re-*

*moving this figure because the description provided in the text is already sufficient.*

Done.

*9. L492 – at this stage, it is not clear why the authors propose a shift of 90 deg. Please homogenize it with the text.*

Next to the VAD representation, the VV90D representation is another way to visualize DL data from a conical scan. As shown in the paper the latter provides a different perspective on the measurement data and reveals characteristic properties of "bad" and "good" estimates, which contributed to the development of the filters presented in the paper. The phase shift of 90 deg has a mathematical background. For DL velocity measurements from conical scans satisfying $V \sim \sin\theta$ a phase shift of 90 deg yields $V90 \sim -\cos\theta$ taking the phase shift identity $\sin(\theta - 90°) = -\cos(\theta)$ into account. Therewith paired data points (x=V90,y=V) plotted in a rectangular coordinate system describe a circle. We describe this in more detail in the revised manuscript in line 284 - 294.

*10. Fig. 12 – Letters [c] and [f] are missing in the caption*

Done. Note that because of the restructuring of the manuscript Fig. 12 in the old version is now Fig. 6 in the new version.

*11. L631 – Please revise "on the one hand".*

Done.

*12. Eq. 9 – Can you please explain the origin of the coefficient 0.6745?*

Here we refer to the work of Iglewicz and Hoaglin (1993).

*13. L744-749 – You can remove this summary of the previous section.*

Done.

**References**

Iglewicz, B. and Hoaglin, D.: How to Detect and Handle Outliers, ASQC basic references in quality control, ASQC Quality Press, 1st edn., 87 pp., ISBN 9780873892476, 1993.

---

## Author Comment (AC2)

**Answer to Referee #2**

**Review of the manuscript amt-2023-153 entitled "Noise filtering options for conically scanning Doppler LiDAR measurements with low pulse accumulation" by E. Paschke and C. Detring**

We would first like to thank the referee for reviewing our manuscript, the overall positive assessment of our work, and for the constructive comments. Below we will respond to the comments and point out changes we made as we revised our manuscript. The reviewer's comments are in black italic; our responses are in blue.

**General Comments**

*The manuscript entitled "Noise filtering options for conically scanning Doppler Lidar measurements with low pulse accumulation" presents the study of a series of post processing methods that can be used to reduce erroneous velocity estimations that are generated by a Doppler lidar. The objective is to increase the measuring accuracy of scanning Doppler lidar in those cases where the scanning speed of the lidar's line-of-sight results in a low laser pulse accumulation. The authors investigate the impact of the increased accuracy of the radial speeds acquired by wind lidar profiler on the estimation of atmospheric turbulence kinetic energy.*

This paragraph calls for a comment to avoid possible misunderstandings. As the generally provided output of the system we use the radial velocity estimates ($V_r$) and the associated signal-to-noise ration (SNR). We work with this output that is made available to the DL user and try to eliminate the incorrect radial velocities contained in the measurement time series. We DO NOT calculate the spectra from the raw data in order to estimate $V_r$ and SNR and do not intend to improve the accuracy of the DL radial velocity estimates. In order to express this more clearly in the manuscript, we have added additional explanatory text in Sect. 2.2 in line 122 - 125

*The authors have a done a good work to present, given the length of the article, a well written and structured manuscript. However, I think that the article is too long. This makes difficult focussing on the differences between the various approaches that are presented in the manuscript and more importantly to identify the message regarding the approach that the authors recommend. For example, the ideas for the new filtering techniques are presented after 22 pages of the manuscript. For this reason, I think that the authors should consider reducing significantly the length of manuscript, by focusing on the new approaches that they implement and test, to improve its readability.*

We understand the criticism about the length of the manuscript. We tried to reduce the length of the manuscript by reorganizing its structure, shortening of individual text passages and reducing the number of illustrations. In particular, significant changes have been made to sections 3 and 5. As part of the restructuring, a stronger focus was sought on the filter methods, which now takes up the main part of the manuscript on pages 12 - 26. The advantages of the newly introduced filtering techniques compared to common filter techniques are then discussed using the example of a special application (TKE retrieval) on pages 27 - 32. Overall, the number of pages in the main part of the manuscript was reduced from 46 pages to 34 pages. At the same time, the number of pages in the appendices increased from 4 to 10.

*Furthermore, the authors state they apply different filtering methods to reduce the noise in Doppler lidars, however I think that rather the noise they reduce the biases of the estimated velocity. A velocity bias in a wind lidar measurement is usually a result of a combination of low atmospheric backscattering and of both the post processing of the Doppler spectra (e.g. noise normalization and thresholding), as well as the frequency estimator of the mean Doppler shift frequency. The latter two parameters are instrument specific. Therefore, I think that it is important to include in the article this information related to the wind lidar used, since they could potential explain the "type B" errors presented in Fig. 1. If it is possible it would be also interesting to show typical Doppler spectra from the "type A" and "type B" errors.*

We feel that the term "noise" is differently understood by the reviewer. When we speak about noise, we mean wrong ("bad") radial velocity estimates (see Sect. 2.2 in the revised manuscript). These "bad" radial velocity estimates can cause errors in the mean wind (i.e. speed and direction) and turbulence variables that we try to retrieve from the DL radial velocity measurements. For that reason we were looking for new filter approaches to eliminate "bad" radial velocity estimates from the measured time series. For the "type B" noise issue we refer to our answer to the reviewers comment on line 139.

**Specific Comments**

*"Section 2. Measurements": for the understanding of the velocity measurements presented in the article it is going to be useful to include information regarding the terrain features around the area where the lidar was located, as well as the local wind climatology.*

Done. (see line 89 - 91 in the revised manuscript)

*Line 139. The authors state that according to their experience random velocity biases are not uniformly distributed in Doppler lidars. I think that this comment should be specific to the instrument used, otherwise please add references here.*

In lines 164 - 173 of the old manuscript we pointed out that the "type B" noise characteristics is not a DL78 specific problem. In the revised manuscript we address this point in line 164 - 175. Additionally we have now supported the statements made here by showing further measurement examples from various DL systems (see Appendix C in the revised manuscript).

*Lines 162 - 163. Is there a noise threshold applied to the Doppler spectra prior the estimation of the Doppler shift in the lidar used in this study? And if yes, is this threshold always the same or does it vary in time? And how are the SNR values presented in Fig. 1 estimated?*

We cannot provide details on that since the analysis of the Doppler spectra is internally performed by the system software and the DL user does not have information on this.

*Lines 166 - 170. The authors write "we can also rule out that the occurrence of type B noise is a systematic DL78 problem. However, couldn't the "type B noise" presented in Fig. 1[i] be a result of RIN noise in the specific Doppler lidar system? If it is not a systematic problem of the DL78 instrument, then what is the origin of this bias?*

Indeed, the occurrence of type B noise seems to be a specific issue of the Streamline DL systems, as is demonstrated in the new Appendix C of the revised manuscript. From a user's perspective we can only state this fact without having the chance to search for the internal origin of it.

*Line 197. I think that there is an error in Eq. (4), according to the way the parameter "C" is defined it should multiply the ACF not divide.*

That's correct. Thanks for the advice. We corrected the mistake. Please see Eqn. B3 in Appendix B of the revised manuscript.

*Page 13. Figure 4, why all the range gates are used in this figure? Shouldn't we expect to see a difference in the features of these plots with height?*

**Answer to the first question:** DL end user usually apply an SNR threshold to be able to differentiate between reliable and non reliable measurements of radial velocity (see also Pearson, (2009)). For practical reasons (and especially for 24/7 routine applications) this threshold must apply equally to all ranges and to different conditions w.r.t to the atmospheric aerosol loading. We therefore plotted measurements from all ranges in order to follow the guidelines of an SNR threshold determination as proposed by Pearson et al. (2009) and Abdelazim (2016). **Answear to the second question:** If it can be assumed that the aerosol loading decreases with altitude, fewer "bad" estimates can generally be expected for measurements at lower ranges than for higher. However, there is additionally a natural variability in the atmospheric aerosol content over the course of a day and such all heights (e.g. depending on the air masses transported with the flow) such that in situations with little aerosol even in lower layers an increased occurrence of "bad" estimates may be observed. For this reason it does not appear practical to define an altitude-dependent SNR threshold.

*Page 19. Figure 9, what does it mean that there is a 43.19% data availability in the unfiltered data? Why isn't it 100%? And how it is possible that the data availability of the data after the application of the CNS filter is higher than the unfiltered ?*

The value 43.19 % refers to "level b" quality-checked data, which is marked with lev b in the plots. For TKE values that fulfill this level the assumptions for the TKE corrections made in Smalikho and Banakh (2017) are fulfilled (see also line 572 - 575 of the revised manuscript). If using unfiltered data with a high proportion of "bad" estimates, these assumptions are often not met so that a large proportion of the derived TKE values does not fulfill the level b criteria in the quality control. Using CNS filtered input data for the TKE retrieval the significantly lower proportion of "bad" estimates leads to better conditions and thus a higher data availability of reliable TKE retrievals. Note that due to the restructuring of the manuscript, Fig. 9 in the old version is now Fig. 16 in the revised version. Additionally Fig. 16 has been modified and does not include TKE values retrieved from unfiltered DL radial velocity measurements any more.

*Line 399. How is it possible to get negative TKE values?*

Note that according to Smalikho and Banakh (2017) turbulence kinetic energy (TKE) is calculated via $TKE = 3/2(\sigma_L^2 + \sigma_t^2 - \sigma_e^2)$ where $\sigma_t^2$ and $\sigma_e^2$ denote correction terms. Negative values are for instance associated with the Smalikho and Banakh (2017) correction for the instrumental error $\sigma_e^2$. If unfiltered radial wind measurements are used to determine this correction, the "bad" estimates they contain can also cause errors in the correction

term, which can ultimately lead to negative TKE values. Note that Fig. 9 has been modified and now been inserted as Fig. 16 in the new manuscript. The modification is that Fig. 16 does not include evaluations of the DL data based on the filter type: "unfiltered". Since the latter is no longer considered in the revised manuscript, there is no need any more to point out possible gaps due to negative TKE values, i.e. this line no longer exists in the revised manuscript.

*Line 515. How is the $\Delta r = 0.5 m/s$ selected? Is it related to the velocity resolution of the Doppler lidar?*

This value turned out to be a viable choice if using the $r_i - count\{V_i\}$ diagram as a tool in the filtering procedure (see line line 303 in the revised manuscript). There could be a connection with the resolution of the Doppler lidar here, but we have not investigated this in more detail.

*Line 658. The use of median absolute deviation has been used in previous studies of filtering random errors from Doppler wind lidars, as in the work of:*
Karagali, I., Mann, J., Dellwik, E., & Vasiljević, N. (2018, June). New European wind atlas: The Østerild balconies experiment. In Journal of Physics: Conference Series (Vol. 1037, No. 5, p. 052029). IOP Publishing.

*Please note that I am neither the author nor one of the co-authors of the above publication.*

Yes, we are aware of that. The filter applied in Karagali et al. (2018) is based on the work of Beck and Kühn (2017). A reference on the latter is given in line 241 of the old manuscript (or line 190 of the revised manuscript).

*Line 661. Eq. (9) where is the number 0.6745 coming from?*

Here we refer to the work of Iglewicz and Hoaglin (1993).

*Line 685. Fig. 17, what is the parameter "g" in the caption of the figure?*
The parameter "g" denotes the standard deviation of "good" estimates (orange colored data points in Fig. 17 [b] of the old manuscript version or in Fig. 11 [a] of the revised version). This value is used to calculate the sine-wave-fit (Smalikho, 2003) into the "good" radial velocity estimates from conically scanning DL measurements (see the black line in Fig. 11 [a]). Note that in both the old and revised manuscript the parameter g is explained in the Figure caption.

*Line 709. The authors write "First, by calculating the previously accepted MIN/MAX radial velocity values for each azimuthal direction.". What happens the first time that the filter is applied where there are no previously accepted min and max values?*

We have expressed ourselves somewhat misleadingly here and reformulated this sentence slightly (see line 485 in the revised manuscript)

*Section 5.1 It is very difficult to visually detect the differences between the full approach I and II in Figs 20 and 21. Given the length of the article the authors could consider removing these figures and keep in the text the key findings of the testing of the two methods.*

We found a compromise here and only removed Fig. 20 because indeed the differences between approach I and II are better seen in the VAD representation (Fig. 21). Note that due to the restructuring of the revised manuscript Fig. 21 in the old version is now Fig. 14 in the revised version.

*Figure 24. The authors present here a comparison between the estimated TKE using the suggested combined filter (CF) approach versus the estimated TKE from a sonic anemometer. The results in respect to RMSD and data availability are comparable to the ones got using the CNS filter. It is claimed that the advantage of the CF is that it gives more systematic results when comparing it to the TKE estimation based on a SNR filter which are presented in Fig. 24 c. The results presented in Fig. 24 c show indeed that there is an improvement in the agreement in the estimated TKE values using the two filtering approaches, however they do not say something whether they are accurate or not.*

In this figure we presented a comparison between the estimated TKE using the SNR filtering technique (x-axis) versus TKE using the CF filtering technique (y-axis). In subfigure 24 [a] this comparison is only for data at 90m height and in Fig. 24 [c] data from measurement heights between 45m and 500m are included. Note that due to the restructuring of the manuscript, Fig. 10 [c],[d] and Fig. 24 [c],[d] in the old manuscript are now combined to Fig. 17 in the revised manuscript. This allows for a better comparison of the retrieved TKE using the CF and CNS filter, respectively. A comparison of the RMSD values (Fig. 17 [a],[c]) and the limits of agreement (Fig. 17 [b],[d]) shows an improvement of TKE values when using the CF filter. We understand the remark "... do not say something whether they are accurate or not" in a way that the reviewer misses a comparison concerning systematic deviations vs. some independent reference data here. In this respect we would like to emphasize that we defined the SNR filtered data as a reference data set here (since we do not have independent TKE

values for altitudes above the tower height) after we had shown earlier that these SNR filtered data (with a lower data availability) compare favourably to the sonic based TKE values at a height of 90m. This is explained in the text in lines 632-638 of the revised manuscript.

*Lines 887 – 889. My hypothesis is that the systematic appearance of close to zero values is probably an issue with either the way that the Doppler spectra are being processed in the cases of low SNR by the StreamLine lidar or an instrument specific technical issue. I think that this should be stated, because they way that it is written right now one could understand as a generic issue of wind lidar profilers.*
Done. See line 680 - 682 in the revised mansucript.

**Technical Corrections**

*I found only few parts that need a technical correction. Please find below a short list of points that either need a clarification or correction.*

*Line 23. What is meant with "all-sky-scanner technique"*
This means that there are no restrictions regarding zenith and azimuth LOS measurement settings (see line 25-26 in the revised manuscript).

*Line 253. I suggest replacing the sentence "is a useful tool to get clear results" with "reduces the variance of the background noise"*
Done. This sentence does not exist anymore in the revised manuscript.

*Line 437. Replace "horizonatl" with "horizontal"*
Done.

*Line 534. Delete one "the" after the "Compared to"*
Done.

*Line 689. Replace "explained with" with "explained by"*
Done.

*Finally, please replace the italic fonts with roman fonts for writing units.*

**References**

Beck, H. and Kühn, M.: Dynamic Data Filtering of Long-Range Doppler Li-DAR Wind Speed Measurements, Remote Sens-Basel, 9, https://doi.org/10.3390/rs9060561, 2017.

Iglewicz, B. and Hoaglin, D.: How to Detect and Handle Outliers, ASQC basic references in quality control, ASQC Quality Press, 1st edn., 87 pp., ISBN 9780873892476, 1993.

Smalikho, I.: Techniques of Wind Vector Estimation from Data Measured with a Scanning Coherent Doppler Lidar, J. Atmos. Ocean. Tech., 20, 276–291, https://doi.org/10.1175/1520-0426(2003)020¡0276:TOWVEF¿2.0.CO;2, 2003.